# CF-VLM: Counterfactual Vision-Language Fine-tuning

**Jusheng Zhang\*[1], Kaitong Cai\*[1], Yijia Fan[1], Jian Wang[2], Keze Wang[1],†**

[1]Sun Yat-sen University
[2]Snap Inc.
†Corresponding author: `kezewang@gmail.com`

## Abstract

Recent advances in vision-language models (VLMs) have greatly improved cross-modal semantic understanding, yet significant limitations remain in fine-grained discrimination and deep causal reasoning tasks. Existing VLMs often rely on superficial statistical correlations, lacking the ability to capture the underlying causal logic between visual and textual content. To address this, we propose CounterFactual Vision-Language Fine-tuning (CF-VLM), a novel framework that enhances the causal reasoning capabilities of VLMs through the targeted use of counterfactual samples. CF-VLM introduces three complementary training objectives: maintaining foundational cross-modal alignment, reinforcing the uniqueness, and stability of factual scene representations against coherent counterfactuals, and sharpening the model's sensitivity to minimal but critical causal edits. Extensive experiments demonstrate that CF-VLM consistently outperforms strong baselines and state-of-the-art methods on compositional reasoning and generalization benchmarks. Furthermore, it shows promise in mitigating visual hallucinations, indicating improved factual consistency. Our CF-VLM provides a robust foundation for deploying VLMs in high-stakes, real-world scenarios requiring reliable reasoning and interpretability code.

## 1 Introduction

With the rapid advancement of vision-language models (VLMs)[1–7] such as CLIP [8], the capability of cross-modal semantic understanding has significantly improved, laying a solid foundation for diverse artificial intelligence applications [1]. However, current VLMs still exhibit fundamental limitations in complex visual-language tasks that demand fine-grained discrimination and deep reasoning[9–16]. These models tend to rely on superficial statistical correlations, struggling to capture the underlying causal logic between visual and textual content[4, 5, 17, 18]. Such limitations are particularly evident in fine-grained recognition and causal reasoning scenarios, constituting a core bottleneck that hampers the applicability of VLMs in high-stakes, real-world environments requiring robust reasoning[19].

To improve fine-grained alignment, recent efforts have extended contrastive learning strategies[20–22], commonly optimizing objectives like the triplet loss $L = \sum [m + S(A, N) - S(A, P)]_+$, where $S(\cdot, \cdot)$ denotes similarity and $m$ is a margin. These methods aim to enforce $S(A, P) > S(A, N) + m$ in the representation space [23], improving discriminative performance. Nevertheless, their primary focus lies in binary distinction—whether pairs match—rather than in understanding the causal attributes that explain how minimal differences in counterfactuals may lead to a semantic shift[5, 24–26]. Consequently, these models often fail to attend to the decisive attributes or relations that govern semantic matching, leading to misinterpretation when faced with counterfactual perturbations.

To overcome these limitations, we propose **CF-VLM** (CounterFactual Vision-Language Fine-tuning), a novel fine-tuning framework designed to enhance the causal reasoning capability of VLMs. Unlike

prior approaches focused solely on improving general discrimination, CF-VLM centers the learning process around *counterfactual samples* [27]. Specifically, *counterfactual samples* are crafted by applying minimal yet semantically crucial edits to an original image or its associated causal relations.

These edits alter the logical meaning of the scene while preserving proximity to the factual base. For instance, changing the color, category, status, or spatial configuration of a core object, or reversing an action-outcome relation, can generate a new counterfactual scenario. Though seemingly subtle, these modifications result in fundamental changes in the semantic match with the accompanying text. CF-VLM thus encourages the model to detect and reason about such *causal decision points*—the critical attributes or relations that determine whether a visual-textual pair truly matches. Building upon cross-modal alignment, CF-VLM introduces two progressive and complementary training objectives to model the contrastive relationships between factual and counterfactual data:

**First**, our CF-VLM reinforces the uniqueness and stability of the *anchor factual scene* $(I_{\text{anchor}}, T_{\text{anchor}})$ by contrasting it with a set of *complete counterfactual scenarios* $(I_{\text{cf}_i}, T_{\text{cf}_i})$, where both image and text are jointly modified to represent a logically consistent yet semantically distinct alternative. The training objective enforces $S(I_{\text{anchor}}, T_{\text{anchor}}) \gg S(I_{\text{cf}_i}, T_{\text{cf}_i})$ for all $i$, thus establishing a clear reference and preventing representation confusion caused by the presence of numerous "parallel realities".

**Second** and more critically, our CF-VLM focuses on enhancing the model's sensitivity to *minimal causal edits*—those that singularly affect the semantic validity of a factual description. In this setting, the factual image $I_{\text{fact}}$ is paired with its original text $T_{\text{fact}}$ (with similarity $S(I_{\text{fact}}, T_{\text{fact}})$), while the same text is also paired with a minimally edited image $I_{\text{cf}}$ (with similarity $S(I_{\text{cf}}, T_{\text{fact}})$). CF-VLM aims to maximize the margin $S(I_{\text{fact}}, T_{\text{fact}}) \gg S(I_{\text{cf}}, T_{\text{fact}})$, thereby guiding the model to focus on the semantic shift caused by critical causal modifications.

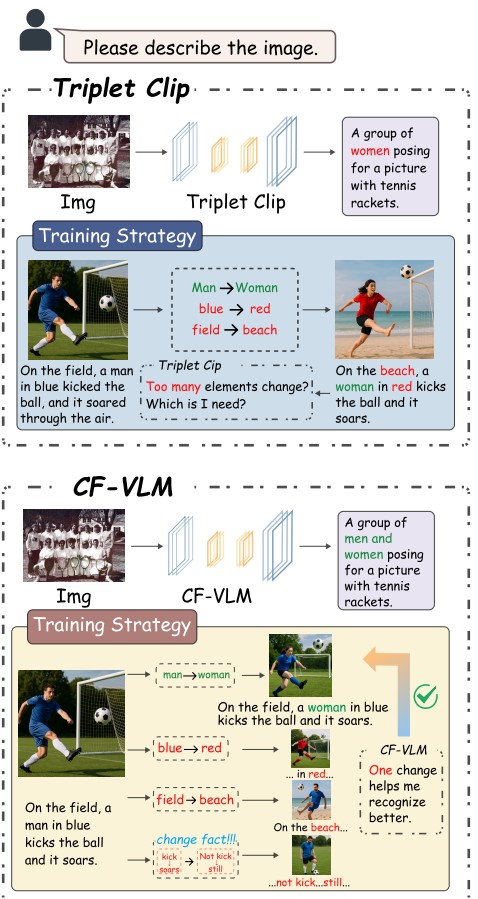

Figure 1: Illustration of CF-VLM's training framework. The model is exposed to both factual and counterfactual image-text pairs, where the latter introduce minimal but semantically decisive edits (e.g., "kick" → "not kick", "man" → "woman"). In contrast to Triplet CLIP, which focuses on coarse-grained similarity (e.g., matching vs. non-matching)

## 2 Preliminary

Before detailing the CF-VLM framework, this section introduces foundational concepts, key notations, and the standard formulation of vision-language models (VLMs) in cross-modal semantic understanding. Representation Learning in Vision-Language Models:The core objective of VLMs is to learn joint multi-modal embeddings that align image and text representations in a shared semantic space[28–30]. Given an image $I$ and its corresponding textual description $T$, a typical VLM consists of two primary encoder components: **Image Encoder** $f_I(\cdot)$: This module maps the input image $I$ to a $d$-dimensional visual embedding vector $\mathbf{e}_I \in \mathbb{R}^d$:$\mathbf{e}_I = f_I(I)$ **Text Encoder** $f_T(\cdot)$: Analogously, the text encoder transforms the input textual sequence $T$ into a $d$-dimensional text embedding vector $\mathbf{e}_T \in \mathbb{R}^d$:$\mathbf{e}_T = f_T(T)$. To ensure the comparability of these embeddings in the metric space and to stabilize similarity computations, both $\mathbf{e}_I$ and $\mathbf{e}_T$ are typically $L_2$-normalized[31] before being used in downstream contrastive learning objectives, resulting in unit-length representations.

## 2.1 Cross-Modal Similarity Measurement

The learned image embedding $\mathbf{e}_I$ and text embedding $\mathbf{e}_T$ reside in a shared representation space[32]. Within this space, semantically aligned image-text pairs are expected to exhibit high similarity, whereas unrelated pairs should demonstrate low similarity. This cross-modal semantic similarity, denoted as $S(I, T)$, is typically measured via the cosine similarity between the normalized embedding vectors, which is mathematically equivalent to their dot product: $S(I, T) = \mathbf{e}_I^\top \mathbf{e}_T = \frac{f_I(I) \cdot f_T(T)}{\|f_I(I)\| \cdot \|f_T(T)\|}$. In practice, since both embedding vectors are L2-normalized[31], the above formulation simplifies to: $S(I, T) = \mathbf{e}_I^\top \mathbf{e}_T$. This similarity score serves as a fundamental computational unit in the contrastive learning objective.

## 2.2 Definition and Types of Counterfactual Samples

Our CF-VLM leverages counterfactual samples for fine-grained supervision. While most existing vision-language learning methods rely on general data augmentation or negative sampling [33], such approaches often lack the capacity to systematically generate the minimal yet semantically critical interventions required for deep causal probing. In contrast, we define a *counterfactual sample* as a new image-text pair, derived from an anchor $(I_a, T_a)$ through a precise, targeted intervention. These edits—guided by prompt engineering for controlled and interpretable manipulation—are specifically designed to enhance the model's understanding of key causal factors underlying image-text alignment. This focus distinguishes our CF-VLM from those methods that merely increase data diversity or construct generic hard negatives. We concentrate on two principal types of counterfactual interventions: (1) **Single Object Attribute Modification:** Targeting a core object in the image, this intervention modifies only one physical attribute at a time—such as color (e.g., red apple → green apple), category (dog → cat), pose or state (standing → lying), spatial relation (left → right), or quantity. The objective is to rigorously test and enhance the model's sensitivity to subtle yet semantically disruptive changes [34]. (2) **Key Causal Relationship Adjustment:** This intervention modifies the key action, event outcome, or causal chain depicted in the scene (e.g., changing "hits the ball" to "misses the ball," or "object falling" to "object floating"), directly challenging the model's understanding of visual dynamics and physical plausibility. Based on these targeted intervention strategies, we construct two categories of counterfactual data structures for training:

- **Complete Counterfactual Scenario Pair** $(I_{cf_k}, T_{cf_k})$: Both the image and the text are jointly edited to form a logically coherent but factually distinct scenario from the anchor, facilitating the learning of plausible alternative realities.

- **Minimally Edited Counterfactual Image** $I_{cf\_edit_j}$: A single critical intervention is applied to $I_a$, and the resulting $I_{cf\_edit_j}$ is paired with the original anchor text $T_a$ as a hard negative sample, enabling fine-grained discrimination.

# 3 Methodology

**CF-VLM Overview:** Our CF-VLM is built upon a pretrained vision-language model and aims to refine its representational space by introducing carefully constructed counterfactual samples along with a set of targeted learning objectives[35–39]. The central insight of CF-VLM is that merely learning to distinguish between matching and non-matching image-text pairs is insufficient. More crucially, the model must develop an understanding of *why* a given pair matches or not—particularly when the differences are subtle and involve causally significant attributes[40]. To this end, the overall training objective of CF-VLM, denoted as $L_{\text{CF-VLM}}$, comprises three key loss components, each responsible for a distinct but complementary aspect of model enhancement: Maintaining the model's foundational cross-modal alignment capability via $L_{\text{align}}$; Strengthening the uniqueness and stability of anchor factual scene representations relative to plausible counterfactuals via $L_{\text{csd}}$; Enhancing the model's causal sensitivity to semantic changes induced by minimal but critical edits via $L_{\text{fcd}}$. These components are integrated through a weighted sum: $L_{\text{CF-VLM}} = \alpha \cdot L_{\text{align}} + \beta \cdot L_{\text{csd}} + \gamma \cdot L_{\text{fcd}}$ where $\alpha$, $\beta$, and $\gamma$ are hyperparameters that balance the contribution of each loss.

## 3.1 Counterfactual Data for Training

The organization of training data plays a pivotal role in fulfilling the learning objectives of CF-VLM. For each training iteration, our CF-VLM is provided with a structured data unit comprising the

following elements: An anchor factual image-text pair $(I_a, T_a)$; a set of $K$ complete counterfactual scenario pairs $(I_{cf_k}, T_{cf_k})k = 1^K$, where each $(Icf_k, T_{cf_k})$ is derived from a targeted modification of $(I_a, T_a)$ and collectively forms a new, internally coherent counterfactual scene; and a set of $J$ minimally edited counterfactual images $I_{cf_edit_j}j = 1^J$, where each $Icf_edit_j$ is generated by applying a single critical semantic edit—such as modifying an object attribute or altering a causal relation—to the anchor image $I_a$. This counterfactual data structure enables CF-VLM to perform targeted supervision across multiple levels of semantic granularity[41–43], ranging from basic alignment to causal sensitivity.

## 3.2 Core Learning Objectives of CF-VLM

**Foundational Cross-Modal Alignment Loss.** To ensure that CF-VLM preserves the foundational image-text alignment capability of the pretrained vision-language model during fine-grained tuning, we incorporate a standard contrastive loss. This loss is consistent with the pretraining objectives used in models such as CLIP and aims to pull semantically relevant image-text pairs closer in the shared embedding space, while pushing apart unrelated pairs. Given a training batch of $N$ factual image-text pairs $\{(I_i, T_i)\}_{i=1}^N$, let $\mathbf{e}_{I_i}$ and $\mathbf{e}_{T_j}$ denote the normalized embeddings of image $I_i$ and text $T_j$, respectively. Their similarity score is computed as: $S(I_i, T_j) = \mathbf{e}_{I_i}^\top \mathbf{e}_{T_j}$ The foundational alignment loss $L_{\text{align}}$ is defined using a symmetric InfoNCE objective:

$$L_{\text{align}} = -\frac{1}{2N} \sum_{i=1}^N \left( \log \frac{\exp(S(I_i, T_i)/\tau)}{\sum_{j=1}^N \exp(S(I_i, T_j)/\tau)} + \log \frac{\exp(S(I_i, T_i)/\tau)}{\sum_{j=1}^N \exp(S(I_j, T_i)/\tau)} \right) \quad (1)$$

Here, $S(I_i, T_j)$ denotes the cosine similarity between the normalized image embedding $\mathbf{e}I_i$ and text embedding $\mathbf{e}T_j$. The first term inside the sum, $-\log \frac{\exp(S(I_i, T_i)/\tau)}{\sum_{j=1}^N \exp(S(I_i, T_j)/\tau)}$, encourages each image $I_i$ to be most similar to its paired text $T_i$, compared to all other texts in the batch. The second term, $-\log \frac{\exp(S(I_i, T_i)/\tau)}{\sum_{j=1}^N \exp(S(I_j, T_i)/\tau)}$, encourages each text $T_i$ to be most similar to its paired image $I_i$, compared to all other images in the batch. The temperature parameter $\tau$ controls the sharpness of the distribution. This symmetric formulation ensures that both image-to-text and text-to-image alignments are jointly optimized.

**Counterfactual Scenario Discrimination Loss.** After establishing basic image-text alignment, CF-VLM introduces the loss term $L_{\text{csd}}$ to reinforce the uniqueness and stability of the representation of the anchor factual scenario. The core idea is that the model should not only understand the current factual instance but also clearly distinguish it from logically coherent yet semantically different *parallel* counterfactual scenarios.

Given an anchor image-text pair $(I_a, T_a)$ and a set of $K$ complete counterfactual scenario pairs $\{(I_{cf_k}, T_{cf_k})\}_{k=1}^K$ derived from it, the $L_{\text{csd}}$ loss enforces that the similarity score $S(I_a, T_a)$ of the anchor pair is significantly higher than that of any counterfactual pair $S(I_{cf_k}, T_{cf_k})$. The loss is implemented as a hinge loss:

$$L_{\text{csd}} = \frac{1}{K} \sum_{k=1}^K \max(0, S(I_{cf_k}, T_{cf_k}) - S(I_a, T_a) + m_1) \quad (2)$$

Here, $S(I_a, T_a)$ denotes the similarity between the anchor image-text pair, while $S(I_{cf_k}, T_{cf_k})$ represents the similarity of the $k$-th counterfactual pair derived from the anchor. For each counterfactual scenario, the loss encourages the similarity of the anchor pair to exceed that of the counterfactual by at least a margin $m_1$. If the counterfactual pair is too close to or even exceeds the anchor in similarity, the hinge loss becomes positive and penalizes the model. When the anchor similarity already surpasses all counterfactuals by at least $m_1$, the loss is zero. This mechanism enforces a clear separation between the factual anchor and its plausible but semantically different counterfactuals in the embedding space, thereby improving the model's discriminative power. **Fine-Grained Causal Discrimination Loss.** The loss term $L_{\text{fcd}}$ represents the core innovation of the CF-VLM framework with respect to causal understanding. It is specifically designed to enhance the model's ability to detect and reason about changes in image-text matching relationships that arise from *minimal but semantically critical edits*, such as single attribute modifications or causal relationship reversals, as defined in Section 2.3. Given an anchor factual image-text pair $(I_a, T_a)$ and a set of $J$ **minimally edited counterfactual images** $\{I_{cf\_edit_j}\}_{j=1}^J$—each generated via an atomic semantic intervention on $I_a$—the objective of $L_{\text{fcd}}$ is to ensure that the similarity score $S(I_a, T_a)$ is significantly higher

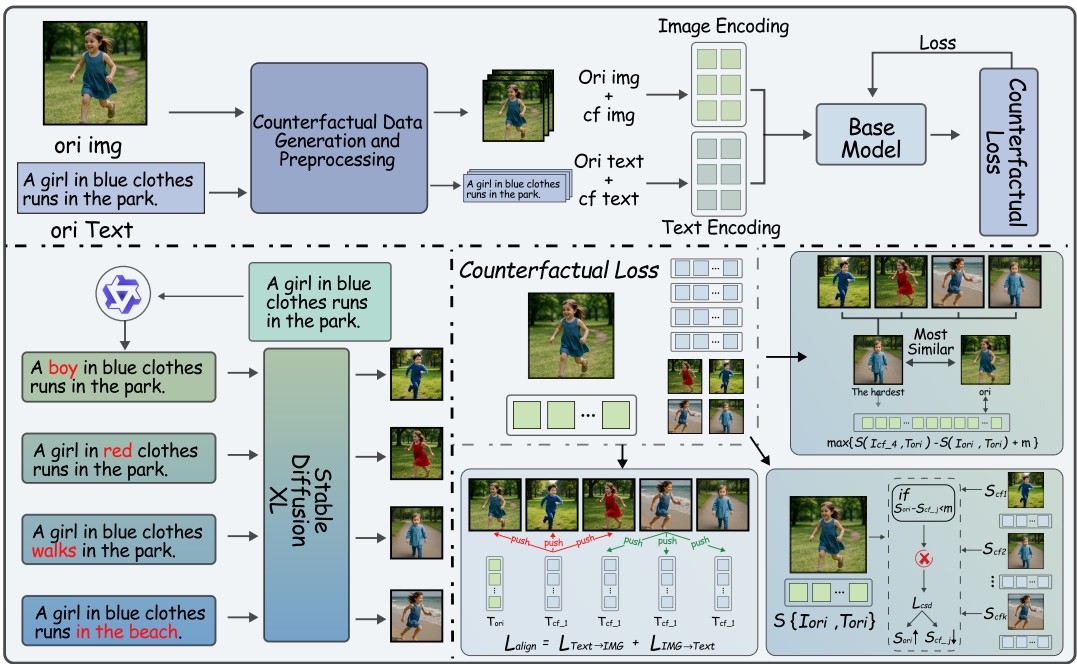

Figure 2: CF-VLM training pipeline. Given a factual image-text anchor, the framework generates complete and minimally edited counterfactual images using a fine-tuned SDXL model. The model is optimized via three complementary objectives: cross-modal alignment ($L_{align}$), counterfactual scenario discrimination ($L_{csd}$), and fine-grained causal discrimination ($L_{fcd}$)—enhancing semantic precision and causal sensitivity.

than that between $T_a$ and any $I_{cf\_edit_j}$. To strengthen training effectiveness, we focus on the most confusing counterfactual (i.e., the one with the highest similarity to $T_a$), thereby forming a *hard negative* sample. The loss adopts a hinge formulation:

$$L_{\text{fcd}} = \max\left(0, \ \max_{j\in\{1..J\}} S(I_{cf\_edit_j}, T_a) - S(I_a, T_a) + m_2\right) \quad (3)$$

Here, $S(I_a, T_a)$ is the similarity between the anchor image and its paired text, while $S(I_{cf_edit_j}, T_a)$ denotes the similarity between the anchor text and the $j$-th minimally edited counterfactual image. Among all counterfactual images, only the one with the highest similarity to $T_a$ (i.e., the hardest negative) is considered in the loss calculation. The hinge loss encourages $S(I_a, T_a)$ to exceed this hardest negative similarity by at least a margin $m_2$; otherwise, the loss penalizes the model. If all counterfactuals are sufficiently less similar than the anchor, the loss is zero. This mechanism ensures the model can robustly detect and distinguish subtle but semantically important changes introduced by minimal edits, thereby enhancing its fine-grained causal reasoning capabilities.

## 4   Experiment

We systematically evaluate our CF-VLM for vision-language models, i.e., Qwen-VL (7B)[44] and CLIP-ViT-B/32, with a focus on enhancing complex visual reasoning. The experimental setup covers pretraining and fine-tuning on filtered CC12M [45] (8.6M image-text pairs) and the CC3M [46] subset (2.6M pairs), with MSCOCO Captions [47] (120K pairs) optionally included for analysis. For each image-text pair, one counterfactual sample is generated by our dynamic counterfactual generation (DCF) strategy, effectively doubling the training data. **Evaluation is conducted on compositional and generalization benchmarks**, including the Conme [48] , ARO [49], VL-Checklist [50], ImageNet-1k [51] (zero-shot classification), and MSCOCO/Flickr30k [52] (zero-shot retrieval). Comparisons include *Zero-shot Qwen-VL (7B)*, *Standard Fine-tuning*, *Text-Negative Fine-tuning*, leading *CLIP-based* models (TripletCLIP [22], CE-CLIP+ [53], COGT-CLIP [54], NegCLIP, Structure-CLIP [55]), *LLM-based VLMs* (LLaVA-1.5 [56], InstructBLIP [57], MiniGPT-4 [58]), and our CF-VLM (Qwen-VL fine-tuned on CC12M+DCF or CC3M+DCF). The counterfactual texts are generated by Qwen2-72B-Instruct with 3-shot chain-of-thought prompting, and images

Table 1: Performance of CLIP-based SOTA vision-language models.

| Method | Replace-Obj | Replace-Attr | Replace-Rel | Conme (Avg) | VG-Rel | VG-Attr | ARO (Avg) | Attribute | Object | Relation | VL-Checklist (Avg) |
|---|---|---|---|---|---|---|---|---|---|---|---|
| NegCLIP (ViT-B/32, CC12M) | 56.9 | 56.7 | 52.2 | 55.27 | 80.9 | 69.9 | 75.4 | 68.3 | 79.2 | 62.8 | 70.1 |
| TripletCLIP (ViT-B/32, CC12M) | 58.2 | 58.2 | 53.2 | 56.53 | 82.6 | 77.4 | 80 | 77.9 | 86.2 | 83.2 | 82.4 |
| CE-CLIP+ (ViT-L/14, CC3M+VG) | 57.5 | 57.9 | 53.8 | 56.4 | 83.6 | 76.2 | 79.9 | 76.3 | 85.2 | 75.3 | 78.9 |
| Structure-CLIP (ViT-B/32, CC12M) | 58.1 | 58.1 | 53.8 | 56.67 | 84.9 | 83.2 | 84.05 | 80.2 | 86.3 | 82.4 | 83.0 |
| COGT-CLIP (ViT-B/32, CC12M) | 58.8 | 60.7 | 54.2 | 57.9 | 87.3 | 88.2 | 87.75 | 84.2 | 81.4 | 86.4 | 84.0 |
| Zero-shot (ViT-B/32) | 50.9 | 51.7 | 46.6 | 49.73 | 59.7 | 61.9 | 60.8 | 68.2 | 81.9 | 63.9 | 71.3 |
| Standard FT (ViT-B/32, CC12M) | 55.7 | 56.4 | 51.2 | 54.43 | 74.2 | 68.2 | 71.2 | 71.4 | 83.1 | 69.4 | 74.6 |
| **CF-VLM (Ours,ViT-B/32, CC12M)** | **60.4** | **60.4** | **56.6** | **59.13** | **88.4** | **90.3** | **89.35** | **88.4** | **87.6** | **89.3** | **88.4** |

Table 2: Performance of LLM-based VLMs, including Qwen-VL and LLaVA variants.

| Method | Replace-Obj | Replace-Attr | Replace-Rel | Conme (Avg) | VG-Rel | VG-Attr | ARO (Avg) | Attribute | Object | Relation | VL-Checklist (Avg) |
|---|---|---|---|---|---|---|---|---|---|---|---|
| InstructBLIP (FlanT5XL, Reported) | 64.1 | 65.2 | 59.9 | 63.07 | 70.5 | 87.4 | 78.95 | 57.2 | 81.3 | 62.4 | 66.97 |
| MiniGPT-4 (7B Vicuna, Reported) | 74.3 | 73.2 | 76.7 | 74.73 | 48.2 | 57.4 | 52.8 | 71.4 | 83.6 | 86.4 | 80.47 |
| Zero-shot Qwen-VL (7B) | 80.7 | 78.4 | 79.4 | 79.5 | 78.3 | 81.4 | 79.85 | 85.6 | 86.4 | 87.9 | 86.63 |
| Standard FT (Qwen-VL, CC12M) | 83.4 | 82.1 | 82.3 | 82.6 | 86.3 | 87.2 | 86.75 | 87.4 | 87.9 | 88.3 | 87.87 |
| TextNeg FT (Qwen-VL, CC12M) | 84.6 | 84.2 | 83.6 | 84.13 | 87.5 | 87.5 | 87.5 | 87.9 | 88.3 | 89.1 | 88.43 |
| **CF-VLM (Ours, Qwen-VL 7B, CC12M)** | **88.3** | **87.5** | **86.9** | **87.57** | **91.8** | **94.6** | **93.2** | **89.6** | **90.7** | **91.4** | **90.57** |
| LLaVA-1.5 (13B, Reported) | 57.6 | 62.4 | 58.6 | 59.53 | 65.4 | 73.2 | 69.3 | 64.3 | 80.6 | 70.2 | 71.7 |
| Standard FT (LLaVA-1.5, CC12M) | 60.8 | 64.1 | 60.4 | 61.77 | 68.4 | 77.4 | 72.9 | 66.2 | 82.4 | 72.3 | 73.63 |
| TextNeg FT (LLaVA-1.5, CC12M) | 61.5 | 65.2 | 62.3 | 63.0 | 69.9 | 78.6 | 74.25 | 67.3 | 83.6 | 74.2 | 75.03 |
| **CF-VLM (Ours, LLaVA-1.5 7B, CC12M)** | **64.4** | **67.8** | **63.7** | **65.3** | **72.3** | **80.3** | **76.3** | **68.6** | **85.1** | **75.6** | **76.43** |

use SDXL 1.0 Base [59] + Refiner (40+15 step schedule). Models are trained using AdamW ($\beta_1 = 0.9, \beta_2 = 0.98, \epsilon = 1 \times 10^{-6}$), a peak learning rate of $1 \times 10^{-5}$ (cosine decay, 500 warmup steps), weight decay 0.1, and batch size 256 on a single NVIDIA A100 (bf16). Training runs for 200K steps (CC12M) or 90K steps (CC3M), with each batch containing a 1:1 ratio of factual and counterfactual samples. The loss is a symmetric triplet contrastive loss $\mathcal{L} = \mathcal{L}_{I \rightarrow T} + \mathcal{L}_{T \rightarrow I}$ with a learnable temperature (init 0.07). **We report main results, ablations, and analyses** to show how counterfactual data aids hallucination mitigation and demonstrate CF-VLM's advantages over SOTA methods. **For evaluation**, compositional reasoning benchmarks use accuracy; ImageNet-1k zero-shot classification is measured by top-1 and top-5 accuracy; image-text retrieval (MSCOCO, Flickr30k) is assessed via Recall@1/5/10. All results are averaged over three random seeds, with standard deviations reported.

## 4.1 Main Results and Analyses

**Compositional Reasoning Performance.** We conduct a comprehensive evaluation of CF-VLM on key compositional reasoning benchmarks, including CONME, ARO, and VL-CHECKLIST, and compare its performance against both baseline models and relevant state-of-the-art (SOTA) approaches. The results are summarized in Table 1 and Table 2. **Performance Gains in CLIP-based Models.** As shown in Table 1, CF-VLM (ViT-B/32, CC12M) outperforms all CLIP-based baselines. On Conme, it achieves 59.13% accuracy, exceeding Std FT by +4.7 points and surpassing TripletCLIP/CE-CLIP+ by +2.6/+2.7 points. On Visual Genome, it obtains 88.4% (VG-Rel) and 90.3% (VG-Attr), outperforming previous models. CF-VLM also achieves 88.4% on the VL-Checklist, showing strong robustness across benchmarks. **Transferability and Advantage in LLM-based Architectures.** As shown in Table 2, CF-VLM delivers strong results with LLM-based models. For Qwen-VL 7B, it achieves 87.57% on Conme, improving over Std FT by +4.9 and over TextNeg FT by +3.4 points. On VG, it reaches 91.8% (Rel) and 94.6% (Attr). CF-VLM also transfers well to LLaVA-1.5, consistently outperforming Std FT and TextNeg FT, and surpasses strong instruction-tuned models like InstructBLIP and MiniGPT-4 across most tasks, demonstrating broad generalization and adaptability.

**Generalization Performance on Standard Vision-Language Tasks.** As shown in Figure 3, CF-VLM achieves strong generalization on standard tasks, including zero-shot ImageNet-1k classification (top-1: 8.83%, top-5: 19.54%) and cross-modal retrieval (R@5: 13.80% on MSCOCO(Avg), 26.58% on Flickr30k(Avg)), outperforming or matching CLIP-based SOTA and relevant baselines.

## 4.2 Ablation Studies

**Effectiveness of Counterfactual Sample Generation.** To assess the impact of counterfactual batches on multi-modal causal reasoning, we design three ablation baselines: (1) **Non-causal Baseline**: training with original pairs and random negatives; (2) **Text-only Counterfactual**: counterfactual

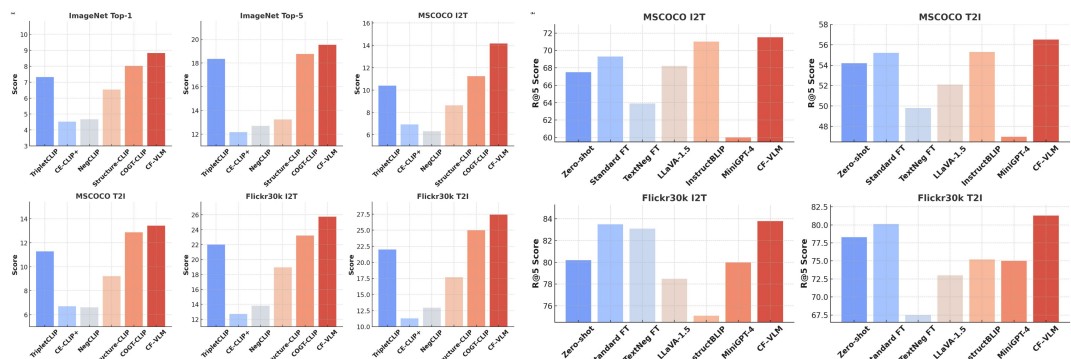

Figure 3: Generalization results. Left (3 columns): CF-VLM boosts baseline performance on CLIP-based classification and retrieval. Right (2 columns): LLM-based evaluation shows improved semantic alignment and Recall@5 over standard finetuning.

Table 3: Performance comparison of different ablation groups on causal reasoning benchmarks. CF-VLM consistently outperforms all baselines across tasks.

| Model | ConMe (Avg) | Replace-Obj | Replace-Attr | Replace-Rel | ARO (Avg) | VG-Rel | VG-Attr |
|---|---|---|---|---|---|---|---|
| Non-causal Baseline | 54.4 | 55.7 | 56.4 | 51.2 | 71.2 | 74.2 | 68.2 |
| Text-only Counterfactuals | 55.9 | 57.3 | 57.6 | 52.8 | 78.65 | 81.9 | 75.4 |
| Image-only Counterfactuals | 56.1 | 57.1 | 57.9 | 53.4 | 78.25 | 82.2 | 74.3 |
| **CF-VLM** | **59.1** | **60.4** | **60.4** | **56.6** | **89.35** | **88.4** | **90.3** |

texts with original images; (3) **Image-only Counterfactual**: counterfactual images with original texts. All experiments use the CF-VLM framework with consistent settings. As shown in Table 3, the full CF-VLM model with both image and text counterfactuals achieves the best results on all benchmarks (e.g., ConMe: 59.1, ARO: 89.35), outperforming all baselines.

Table 4: Ablation study of different loss components on the ConMe dataset. The full model with all three loss terms achieves the best average performance.

| Foundational Cross-Modal Alignment Loss | Counterfactual Scenario Discrimination Loss | Fine-Grained Causal Discrimination Loss | Replace-Obj | Replace-Att | Replace-Rel | Avg |
|---|---|---|---|---|---|---|
| ✓ | ✓ | ✗ | 59.4 | 59.2 | 55.1 | 57.9 |
| ✗ | ✓ | ✓ | 58.9 | 58.1 | 54.8 | 57.3 |
| ✓ | ✗ | ✓ | 59.3 | 59.4 | 55.4 | 58.0 |
| ✓ | ✗ | ✗ | 57.4 | 57.8 | 53.5 | 56.2 |
| ✗ | ✓ | ✗ | 56.8 | 56.8 | 52.9 | 55.5 |
| ✗ | ✗ | ✓ | 56.6 | 56.4 | 53.1 | 55.4 |
| ✓ | ✓ | ✓ | **60.4** | **60.4** | **56.6** | **59.1** |

### Effectiveness of Counterfactual Causal CLIP Losses

To evaluate the impact of different loss components, we perform an ablation study on ConMe using: (1) **Foundational Cross-Modal Alignment Loss**, (2) **Counterfactual Scenario Discrimination Loss**, and (3) **Fine-Grained Causal Discrimination Loss**. We test various configurations by adding or omitting each loss. The full model with all three losses achieves the best average accuracy (59.1), with 60.4 on *Replace-Obj*, 60.4 on *Replace-Attr*, and 56.6 on *Replace-Rel*. Removing any loss reduces performance, e.g., excluding fine-grained causal discrimination drops *Replace-Rel* by 1.5 points (average 57.9). These results demonstrate that combining alignment, counterfactual, and causal objectives is crucial for compositional reasoning in multi-modal models.

### 4.3 Effect of Counterfactual Supervision on Compositional Generalization

**Impact of Counterfactual Types on Task-specific Accuracy.** To further investigate the source of performance gains, we analyze the contribution of different types and quantities of counterfactual samples. The left of Figure 4 reports the results of selectively removing specific types of

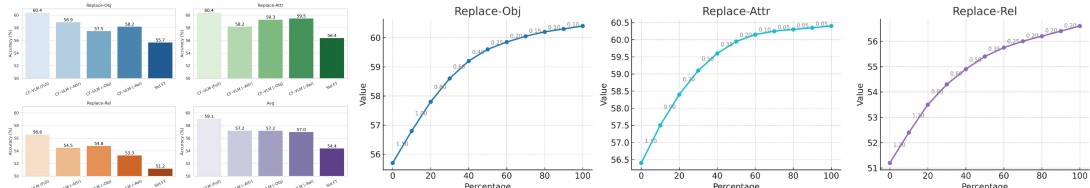

Figure 4: Removing any counterfactual type hurts task accuracy (left); more counterfactuals improve performance with diminishing returns (right).

counterfactual examples—namely, attribute-level (Attr), object-level (Obj), and relation-level (Rel) modifications—from CF-VLM trained with ViT-B/32 on the ConMe benchmark.

The results indicate that omitting any single category of counterfactuals leads to a noticeable performance drop on the corresponding subtask. For instance, removing attribute-level counterfactuals reduces accuracy on the *Replace-Attr* task by 2.2 points (from 60.4 to 58.2), while removing object- and relation-level samples results in decreases of 2.9 and 3.3 points on *Replace-Obj* and *Replace-Rel*, respectively. The overall average accuracy also declines accordingly.

**Effect of Counterfactual Quantity: Tradeoff Between Performance and Efficiency.** We further examine the effect of counterfactual data quantity on model performance. As shown in the right of Figure 4, the average accuracy on the ConMe benchmark increases steadily as the proportion of counterfactual samples (relative to the number of original positive samples) grows—from 0% (Standard Fine-tuning, 54.43%) to 100% (CF-VLM, 59.13%).

The performance curve begins to plateau between approximately 60% and 80% counterfactual ratio, suggesting diminishing marginal gains as more counterfactual data is introduced. This observation highlights a practical tradeoff: in real-world scenarios, one can balance performance improvement and computational cost by selecting an appropriate volume of counterfactual supervision.

## 4.4 Object-Level Hallucination Evaluation

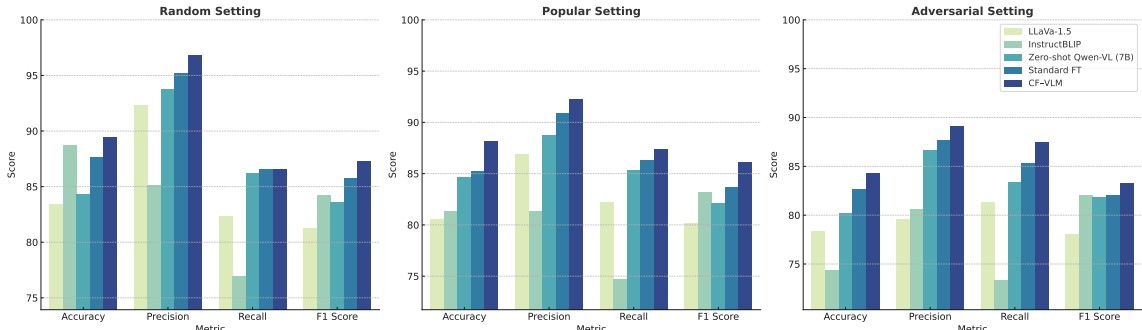

Figure 5: Hallucination evaluation on the POPE benchmark. CF-VLM improves accuracy, precision, recall, and F1 on the object existence task over baselines.

To further assess the practical applicability of CF-VLM, we evaluate its potential in mitigating visual hallucinations—a long-standing challenge for vision-language models. Although CF-VLM is not explicitly designed to optimize hallucination suppression, we hypothesize that its enhanced capability for fine-grained semantic modeling may indirectly improve factual alignment. To validate this hypothesis, we conduct evaluations on two widely adopted hallucination benchmarks: POPE [60] and MME [61].

As shown in Figure 5, on POPE's object existence task, *Standard FT* (Qwen-VL, CC12M) significantly reduces the false positive rate compared to the zero-shot baseline, demonstrating the factual benefits of supervised fine-tuning. Building upon this, *CF-VLM* achieves marginal improvements across all four metrics—accuracy, precision, recall, and F1 score—with average gains of 0.5–1.2 percentage points.

MME benchmark results (Figure 6) show CF-VLM outperforming Standard FT on hallucination-sensitive subtasks (*Existence*, *Count*, *Position*, *Color*). Notable gains in *Color* (+2.0) and *Position*

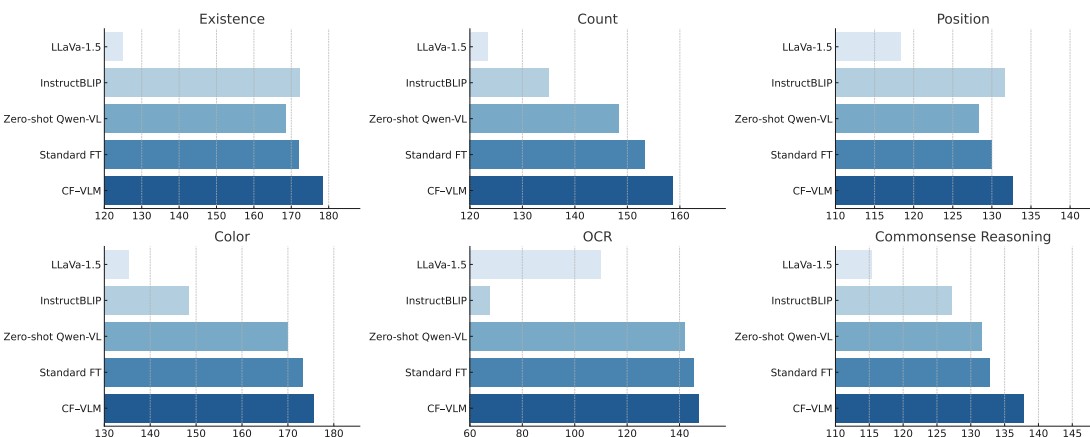

Figure 6: Hallucination evaluation on the MME benchmark. Relative performance gains on MME show CF-VLM's advantage in hallucination-sensitive tasks, especially for color and position, indicating enhanced semantic grounding via counterfactual training.

(+1.4) highlight counterfactual supervision's benefit for attribute-level reasoning, likely due to minimal perturbations in attribute representations reducing hallucination.

## 5   Related Work

Recent advances in vision-language models (VLMs)[62–64], such as CLIP [8] and BLIP [1], have demonstrated remarkable performance in cross-modal tasks through large-scale image-text pretraining. However, these models often struggle with fine-grained discrimination and causal reasoning, relying on superficial statistical correlations rather than capturing deep causal relationships between visual and textual content [5, 10]. Existing approaches employ contrastive learning to enhance embedding space discriminability, such as triplet loss[23], but primarily focus on binary matching, overlooking the impact of causal attribute changes on semantics [25].

Counterfactual learning has emerged as a promising approach to address the causal reasoning limitations of VLMs[24, 65, 66]. introduced counterfactual visual explanations by modifying key visual elements to analyze model decisions, while CPL [25] leveraged counterfactual prompts to improve cross-modal understanding. However, these methods typically focus on single-modality counterfactuals, lacking joint image-text counterfactual designs [67] and using singular objectives that fail to leverage counterfactual samples for multi-level semantic supervision.

Our CF-VLM framework advances VLM causal reasoning by integrating joint image-text counterfactual samples with three complementary training objectives: cross-modal alignment, counterfactual scenario discrimination, and fine-grained causal discrimination. Unlike TripletCLIP [22], which optimizes coarse-grained similarity, CF-VLM targets causal decision points, achieving superior accuracy on benchmarks like ARO, ConMe, and VL-Checklist, while reducing visual hallucinations, demonstrating enhanced robustness and applicability in high-stakes scenarios.

## 6   Conclusion and Limitations

We propose CF-VLM, a counterfactual fine-tuning framework to enhance semantic granularity and causal reasoning in vision-language models (VLMs). Addressing limitations in distinguishing subtle semantic differences, CF-VLM uses minimally edited image-text pairs from a fine-tuned SDXL pipeline, introducing counterfactual scenario discrimination and fine-grained causal discrimination objectives. Evaluations on compositional reasoning (ARO, ConMe, VL-Checklist) and hallucination benchmarks (POPE, MME) show CF-VLM outperforming CLIP-based and LLM-augmented VLMs, improving factual grounding and reducing hallucinations, even on untrained benchmarks, demonstrating the generality of counterfactual supervision. Future work includes integrating human-in-the-loop editing for richer counterfactuals, applying CF-VLM to tasks like VQA or image editing, and exploring interpretability via counterfactual sensitivity for more transparent VLMs.

## Acknowledgements

This work was supported in part by the National Natural Science Foundation of China (NSFC) under Grant 62276283, in part by the China Meteorological Administration's Science and Technology Project under Grant CMAJBGS202517, in part by Guangdong Basic and Applied Basic Research Foundation under Grant 2023A1515012985, in part by Guangdong-Hong Kong-Macao Greater Bay Area Meteorological Technology Collaborative Research Project under Grant GHMA2024Z04, in part by Fundamental Research Funds for the Central Universities, Sun Yat-sen University under Grant 23hytd006, and in part by Guangdong Provincial High-Level Young Talent Program under Grant RL2024-151-2-11.

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

# A    Dataset Details

Our training and evaluation are conducted on a series of standardized image-text datasets and structured evaluation benchmarks, as detailed below:

- **Training Data:** We utilize the cleaned version of CC12M (8.6M image-text pairs) and its subset CC3M (2.6M pairs) as our primary training corpora. All samples undergo standardized preprocessing, including image resizing to $224 \times 224$, text normalization, and language alignment to ensure stable input distribution.

- **Counterfactual Sample Generation:** For each original image-text pair, we generate a semantically similar but minimally perturbed counterfactual pair using our Dynamic Counterfactual Generation (DCF) strategy. This effectively doubles the training data and enhances the model's capacity to learn fine-grained semantic variations through contrastive supervision.

- **MSCOCO Captions (120K pairs):** This dataset is not used for training, but is optionally included for auxiliary analysis tasks (e.g., image-text retrieval or structural generalization), to assess the transferability of our method beyond the core training corpus.

- **Evaluation Benchmarks:** We evaluate the model's compositional reasoning and cross-modal generalization on the following benchmarks:

  - **ConMe Suite:** This benchmark focuses on the model's ability to distinguish subtle semantic changes in compositional logic, such as negation, sequence, and conjunction. Each sample presents an image with two candidate captions—one semantically correct and one minimally perturbed. The task evaluates causal reasoning and structural understanding.

  - **ARO (Attribute–Relation–Object):** This benchmark introduces targeted perturbations to image-text pairs along three dimensions: attributes, relationships, and object identity. It assesses the model's consistency in recognizing multi-dimensional semantic alignment, particularly in fine-grained visual-linguistic scenarios.

  - **VL-Checklist:** This benchmark contains three sub-tasks—Replace-Obj, Replace-Attr, and Replace-Rel—where only one semantic element in the text is systematically modified. It is designed to evaluate the model's sensitivity to attribute- and relation-level counterfactuals.

  - **ImageNet-1k (Zero-shot Classification):** This task transforms the standard ImageNet classification problem into an image-text matching setup. Each image is paired with multiple candidate textual descriptions, and the model must select the most semantically aligned one. This evaluates the model's ability to generalize to natural image distributions in a zero-shot setting.

  - **MSCOCO / Flickr30k (Image-Text Retrieval):** These retrieval tasks involve bi-directional matching (image → text and text → image). We report Recall@1 and Recall@5 metrics to evaluate whether the model can accurately locate the correct match from a large candidate pool.

- **Preprocessing and Splits:** All datasets follow their official train/validation/test splits. During training, each batch maintains a fixed 1:4 ratio between factual and counterfactual samples, to enhance the model's sensitivity to semantically critical perturbations.

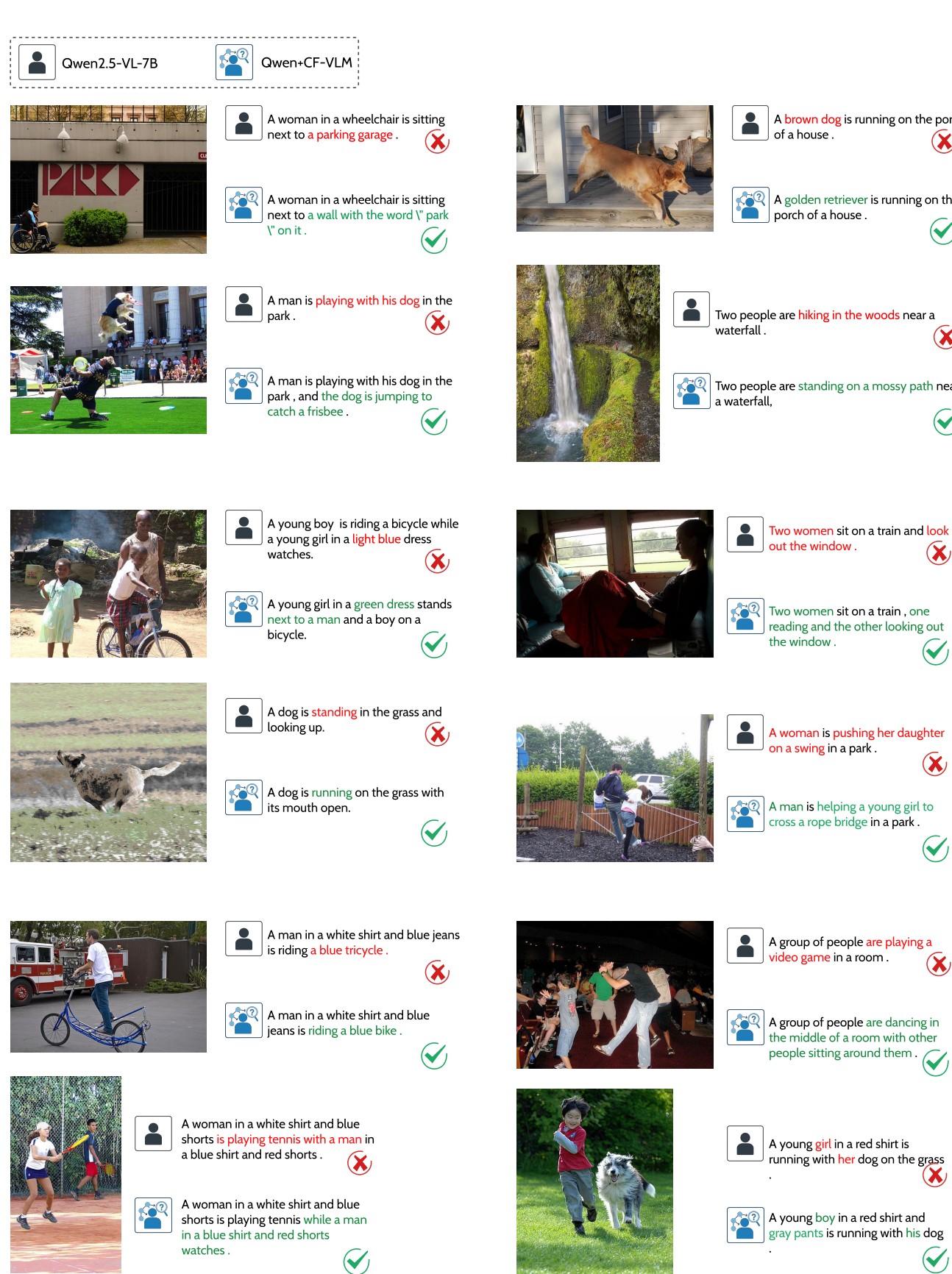

Figure 7: Qualitative comparisons showing CF-VLM's improvements in fine-grained attribute recognition, causal action modeling, and relational understanding across diverse visual scenes.

## Counterfactual Rewriting Prompt

You are a rewriting engine specialized in generating counterfactual or counter-attribute descriptions.

**Your task:** Rewrite the input sentence by changing **exactly one element** — either:

- a **physical attribute of an object**, or
- a **causal relationship**.

**You may change one of the following object attributes (be creative and visually expressive):**

- Color (e.g., red → blue)
- Material (e.g., wood → metal)
- Shape (e.g., round → square)
- Size (e.g., large → small)
- Position or orientation (e.g., on the left → on the right)
- State or pose (e.g., standing → lying)
- Object category (e.g., dog → cat)
- Parts or structure (e.g., three legs → four legs)
- Movement status (e.g., jumping → still)
- Quantity (e.g., two → one)
- Facial expression or emotion (e.g., smiling → angry)

*You may use imaginative or creative combinations as long as they are visually descriptive and logically coherent.*

**Alternatively, you may flip a causal relationship:**

1. Identify a cause-effect link in the sentence.
2. Invert it: If cause A didn't happen, effect B would change.
3. Rewrite the sentence naturally under the new causal logic.

**Rules:**

- Modify **only one thing** (either one attribute or one causal link).
- Keep all other elements in the sentence **exactly the same**.
- Do **not** repeat any previously generated sentence.
- The output must be **a single fluent, grammatically correct sentence**.
- **Do not** include any explanation, comment, tag, or prefix.
- **Only** return the rewritten sentence. Nothing else.

**Format:** Please output the *K* counterfactual sentences in a numbered list:

1. [sentence #1]

2. [sentence #2]

3. [sentence #3]

...

**Example Input:**
*A young woman holding a racket hit the ball, and the ball flew outward.*
**Output (attribute):**

1. An old woman holding a racket hit the ball, and the ball flew outward.
2. A transparent woman holding a racket hit the ball, and the ball flew outward.

**Output (causal):**

3. A young woman holding a racket missed the ball, and the ball dropped to the ground.

**Now return *K* distinct counterfactual rewrites in the required format.**

# B Qualitative Analysis of Post-training Performance

## B.1 Diversity and Quality Control of Counterfactual Samples

CF-VLM employs a structured counterfactual editing strategy that includes four types of controlled interventions: attribute substitution, object substitution, relationship substitution, and location substitution. As illustrated in Figure 8, attribute edits (e.g., replacing "dirt" with "paved roads") and object edits (e.g., changing "two men" to "two women") enable the model to learn from variations in properties and subject identities. Meanwhile, relationship edits (e.g., switching from "throwing a pitch" to "catching a ball") and location edits (e.g., changing the spatial configuration between a "cat" and a "television") enhance the model's understanding of action sequences and spatial semantics.

These controlled counterfactuals enable CF-VLM to cover a wider range of semantic types—from color and texture to interaction and spatial reasoning—while avoiding issues commonly observed in large language model (LLM)-based generation, such as redundancy and logical inconsistency. This structured supervision improves both the semantic diversity and logical coherence of the training samples.

Figure 7 provides qualitative evidence supporting this claim. In the top-left example, the baseline model incorrectly identifies the scene as "a parking garage" and ignores the textual cue "park" on the wall, while CF-VLM accurately describes the scene with attribute-aware recognition. In the second-left row, CF-VLM enhances the original caption "a man is playing with his dog" by adding "and the dog is jumping to catch a frisbee," capturing the entire action-outcome chain. Similarly, in the top-right image, the baseline produces a generic "brown dog," while CF-VLM correctly identifies it as a "golden retriever," offering greater specificity. In another example, CF-VLM replaces "hiking near a waterfall" with "standing on a mossy path near a waterfall," yielding more grounded spatial reasoning. These examples demonstrate CF-VLM's clear advantage in fine-grained attribute recognition, action completion, and relationship understanding. The generated counterfactuals are both more diverse and semantically consistent than those produced by conventional methods.

## B.2 Causal Depth

CF-VLM also demonstrates superior causal reasoning by explicitly modeling the structural components of visual events, including agents, objects, action stages, and outcome states. During training, counterfactual interventions are applied in a structured manner, enabling the model to learn coherent "intervention-effect" causal chains.

As shown in Figure 7, CF-VLM captures rich event semantics across a range of complex scenes. In the "man-dog-frisbee" scenario, CF-VLM identifies a three-step causal sequence: "a man is playing with his dog" → "the dog is jumping to catch a frisbee," exhibiting strong temporal and causal coherence. In contrast, the baseline merely describes surface-level actions without modeling the outcome.

In the "tennis" example, CF-VLM accurately distinguishes between the active player and the observer, generating the caption "a woman is playing tennis while a man watches," thus capturing role-based causal dynamics. In other scenes—such as helping a child cross a bridge or shifting spatial relationships—CF-VLM successfully tracks pre- and post-action semantic states, encoding progression and continuity within the event timeline.

These results indicate that CF-VLM not only improves action recognition but also accurately models temporal structures and agent-object interactions. Overall, CF-VLM outperforms conventional baselines in understanding complex visual events, capturing semantically consistent changes triggered by causal interventions. This structured causal modeling provides a stronger foundation for generalization and discrimination in multimodal semantic tasks, validating the effectiveness of counterfactual supervision for compositional reasoning.

# C Detailed Comparison with Related Work and Highlighting Unique Contributions

In order to clarify CF-VLM's unique contributions in enhancing fine-grained discrimination and advancing deeper causal understanding in vision-language models (VLMs), and to directly address

concerns about novelty, this section presents a thorough comparison of CF-VLM against several key related works at the core mechanism level. We aim to demonstrate that CF-VLM is not a simple application of counterfactual samples but a carefully designed, multifaceted framework to improve sensitivity to specific interventions and identify causal visual elements.

## C.1 Mechanistic-Level Comparison

We compare CF-VLM and representative works—including TripletCLIP, CPL [68], Goyal et al. [69], and Rao et al. [70]—along four dimensions: the type and generation of counterfactual samples, core learning objective/loss design, focus on causality or fine-grained discrimination, and simultaneous handling of image and text counterfactuals.

**Type and Generation of Counterfactual Samples**   CF-VLM adopts a comprehensive strategy for constructing counterfactuals: it combines *complete counterfactual scenarios* where both image and text are jointly edited, with *minimally edited* image perturbations targeting single object attributes or causal relations (e.g., color, class, state). We use fine-tuned SDXL 1.0 for image generation and Qwen2-72B-Instruct for text editing to ensure semantic control.

In contrast:

- **TripletCLIP[22]** synthesizes visual–language negatives via rule- or template-based text perturbations to improve compositional reasoning.
- **CPL [68]** generates counterfactuals by modifying textual prompts using predefined templates to enhance cross-modal robustness.
- **Goyal et al. [69]** focuses on image-based edits of key elements for interpretability, not directly optimizing prediction performance.
- **Rao et al. [70]** learns counterfactual attention by masking or replacing salient regions, calibrating attention maps for visual tasks.

**Core Learning Objectives and Loss Design**   CF-VLM's training objective integrates three complementary losses:

1. **Cross-modal Alignment Loss** $L_{\mathrm{align}}$ maintains baseline image–text matching and prevents forgetting.
2. **Counterfactual Scene Distinction Loss** $L_{\mathrm{csd}}$ contrasts factual versus complete counterfactual scenarios to reinforce representational uniqueness.
3. **Fine-grained Causal Distinction Loss** $L_{\mathrm{fcd}}$ sharpens sensitivity to minimal but critical causal edits.

Other works typically use a single or simpler combination of objectives: TripletCLIP employs a standard triplet loss; CPL uses contrastive learning on textual prompts; Goyal et al. and Rao et al. use counterfactuals primarily for interpretability or attention robustness rather than end-to-end optimization.

**Focus on Causality and Fine-Grained Discrimination**   CF-VLM explicitly targets sensitivity to controlled local interventions ("causal decision points") that determine image–text matching. Unlike TripletCLIP's focus on compositional negative mining or CPL's OOD robustness objective, CF-VLM trains the model to understand why subtle changes alter semantics. Goyal et al. emphasizes interpretability and Rao et al. enhances attention robustness but neither systemically models causal edits tied to fine-grained semantic shifts.**Simultaneous Handling of Image and Text Counterfactuals** CF-VLM uniquely processes both jointly edited image–text scenarios in $L_{\mathrm{csd}}$ and pairs minimally edited images with original text in $L_{\mathrm{fcd}}$. TripletCLIP and CPL focus on text-only counterfactuals, while Goyal et al. and Rao et al. operate primarily on image perturbations. CF-VLM's dual-modality approach provides richer, multi-level supervisory signals.

## C.2 Analysis of CF-VLM's Uniqueness and Integrative Advantages

CF-VLM's distinctive integrative strengths can be summarized as:

1. **Diverse, Controlled Counterfactuals**: Combines high-quality minimal-edits and complete scenarios to offer multi-level contrast.

2. **Multi-Objective Synergy**: $L_{\mathrm{align}}$, $L_{\mathrm{csd}}$, and $L_{\mathrm{fcd}}$ work together to maintain alignment, distinguish scenes, and detect causal edits.

3. **Deep Focus on Causal Decision Points**: Guides the model to learn why small edits affect semantics, advancing causal understanding beyond feature matching.

4. **End-to-End, Controllable Fine-Tuning Framework**: Integrates SDXL and LLM-based generation into a deployable pipeline to retrofit existing VLMs.

## D   Detailed Experimental Setup and Hyperparameters

This section provides a comprehensive overview of the experimental setup, including hardware and software configurations, dataset preparation specifics, and detailed hyperparameter settings used for training and evaluating our proposed **CF-VLM** framework and all baseline models.

### D.1   General Setup

All experiments were conducted on a consistent hardware and software environment to ensure fair comparisons.

**Hardware:** Experiments were primarily run on NVIDIA A100 GPUs. Depending on model size and batch configuration, between 1 to 8 GPUs were utilized per experimental run. **Software:** Key software libraries included Python 3.9+, PyTorch 1.13.1 (with CUDA 11.7), Transformers 4.28.1, and a customized fork of OpenCLIP for certain baseline implementations. Standard scientific computing libraries such as NumPy 1.23.5 and Pandas 1.5.3 were used for data handling and analysis. **Operating System:** All nodes ran Ubuntu 20.04 LTS.

### D.2   Dataset Preprocessing and Splits

Datasets including CC12M, CC3M, MSCOCO Captions, ARO, Conme, VL-Checklist, ImageNet-1k, and Flickr30k were used. For CC12M and CC3M, we followed standard filtering protocols similar to those described in prior large-scale VLM pretraining works, removing images with insufficient resolution, near-duplicate captions, and potential NSFW content based on automated filters. For pretraining/fine-tuning on CC12M/CC3M, we used a 99% training and 1% validation split, randomly sampled. For evaluation on downstream tasks, we adhered to their standard publicly available splits. For ImageNet-1k, standard validation set images were used for zero-shot classification. For MSCOCO and Flickr30k, we utilized the Karpathy splits for zero-shot retrieval.

### D.3   Hyperparameter Settings for CF-VLM Fine-tuning

Our CF-VLM framework was fine-tuned on pretrained Qwen2.5-VL (7B) and LLaVA-1.5 (7B/13B variants where applicable) models. The core hyperparameters for CF-VLM fine-tuning are detailed in Table 5.

The selection of loss weights $(\alpha, \beta, \gamma)$ and margins $(m_1, m_2)$ was performed via grid search on a small subset of the CC3M validation data, optimizing for average performance Improvement on key compositional reasoning benchmarks (ARO, Conme-Avg). The temperature $\tau$ for $L_{\mathrm{align}}$ was generally kept consistent with standard CLIP pretraining practices or the original VLM's configuration.

### D.4   Baseline Model Hyperparameters

For all baseline models (e.g., Zero-shot, Standard Fine-tuning, Text-Negative Fine-tuning, Triplet-CLIP*, CE-CLIP+, etc.), we endeavored to reproduce their results using publicly available codebases and reported hyperparameters where available. For our own Standard Fine-tuning (Std FT) and Text-Negative Fine-tuning (TextNeg FT) baselines applied to Qwen-VL and LLaVA-1.5, we used a setup similar to CF-VLM's $L_{\mathrm{align}}$ component, with identical optimizer settings, learning rates, and batch sizes to ensure fair comparison. Key differences are outlined below:

Table 5: Core hyperparameter settings for CF-VLM fine-tuning. Values were kept consistent across different base models (Qwen-VL, LLaVA-1.5) unless specified. Slight variations (e.g., in learning rate or batch size) were explored during initial tuning, with the reported values yielding optimal performance on a held-out validation set derived from CC3M.

| Hyperparameter | Value / Range |
|---|---|
| Optimizer | AdamW |
| AdamW $\beta_1$ | 0.9 |
| AdamW $\beta_2$ | 0.98 (Qwen-VL), 0.985 (LLaVA-1.5) |
| AdamW $\epsilon$ | $1 \times 10^{-6}$ (Qwen-VL), $1 \times 10^{-7}$ (LLaVA-1.5) |
| Peak Learning Rate ($lr_{peak}$) | $1 \times 10^{-5}$ (for 7B models), $8 \times 10^{-6}$ (for 13B models) |
| Learning Rate Schedule | Cosine decay with linear warmup |
| Warmup Steps | 500 (CC12M), 300 (CC3M) |
| Weight Decay | 0.1 (Qwen-VL), 0.08 (LLaVA-1.5) |
| Effective Batch Size | 256 (distributed across GPUs) |
| Micro-batch Size per GPU | Varied (4 to 16, depending on GPU memory and model) |
| Gradient Accumulation Steps | Adjusted to achieve effective batch size |
| Training Steps (CC12M+DCF) | 200,000 |
| Training Steps (CC3M+DCF) | 90,000 |
| Mixed Precision | BF16 |
| Gradient Clipping Norm | 1.0 |
| Dropout (where applicable in VLM head) | 0.1 |
| Image Resolution | 224x224 (CLIP-based), 336x336 or 448x448 (for Qwen-VL/LLaVA, model dependent) |
| **CF-VLM Specific Loss Parameters** | |
| $L_{\text{align}}$ Temperature ($\tau$) | 0.07 (initial, learnable for some CLIP backbones, fixed for LLM-based VLMs) |
| $L_{\text{csd}}$ Margin ($m_1$) | 0.25 |
| $L_{\text{fcd}}$ Margin ($m_2$) | 0.30 |
| Loss Weight $\alpha$ (for $L_{\text{align}}$) | 1.0 |
| Loss Weight $\beta$ (for $L_{\text{csd}}$) | 0.45 |
| Loss Weight $\gamma$ (for $L_{\text{fcd}}$) | 0.55 |
| Counterfactual Sample Ratio (in batch) | 1:1 (Factual : Counterfactual) |

- **Zero-shot Baselines:** Evaluated directly using the official pretrained checkpoints without any fine-tuning.

- **Standard Fine-tuning (Std FT):** Used only the $L_{\text{align}}$ loss component with factual image-text pairs from CC12M or CC3M. Hyperparameters mirrored those in Table 5 for optimizer, LR, batch size, etc.

- **Text-Negative Fine-tuning (TextNeg FT):** Similar to Std FT, but for each factual pair $(I, T)$, a hard text negative $T_{neg}$ (randomly sampled from other texts in the batch, or generated via simple rule-based negation for some experiments) was used to augment the contrastive loss. The negative sampling strategy and loss formulation followed common practices in the literature.

- **Other Reported Baselines (e.g., TripletCLIP*, COGT-CLIP):** We report scores directly from the respective publications or their official repositories. If re-running was necessary for a specific component (e.g., CLIP-ViT-B/32 on CC12M), we followed their reported settings as closely as possible. For instance, TripletCLIP* variants often use a margin of 0.2 for their triplet loss. NegCLIP typically involves in-batch negatives or a more sophisticated negative cache.

A summary of key training parameters for our implemented baselines is presented in Table 6.

## D.5 Counterfactual Sample Generation Parameters

The generation of counterfactual text descriptions was performed by LLaMA-3-70B-Instruct using 3-shot chain-of-thought prompting. Counterfactual images were generated using SDXL 1.0 Base + Refiner with a 40+15 step schedule. For each factual image-text pair, four counterfactual samples (either image or text modified, or both for complete counterfactual scenarios) were generated, maintaining a 1:4 ratio of factual to counterfactual data during CF-VLM training stages that utilize them.

Table 6: Key hyperparameters for our implemented baseline fine-tuning experiments (Std FT, TextNeg FT). These largely mirror the CF-VLM settings for core optimizer and learning rate parameters to ensure comparability.

| Hyperparameter | Std FT (Qwen-VL/LLaVA) | TextNeg FT (Qwen-VL/LLaVA) |
|---|---|---|
| Base Model | Qwen-VL (7B), LLaVA-1.5 (7B/13B) | Qwen-VL (7B), LLaVA-1.5 (7B/13B) |
| Optimizer | AdamW | AdamW |
| AdamW $\beta_1, \beta_2, \epsilon$ | Same as CF-VLM (Table 5) | Same as CF-VLM (Table 5) |
| Peak Learning Rate | Same as CF-VLM | Same as CF-VLM |
| LR Schedule | Cosine decay with warmup | Cosine decay with warmup |
| Warmup Steps | Same as CF-VLM | Same as CF-VLM |
| Weight Decay | Same as CF-VLM | Same as CF-VLM |
| Effective Batch Size | 256 | 256 |
| Training Data | CC12M or CC3M (Factual pairs only) | CC12M or CC3M (Factual + Text Negatives) |
| Loss Function | Symmetric InfoNCE ($L_{align}$) | Symmetric InfoNCE with hard negatives |
| Contrastive Temperature ($\tau$) | 0.07 (fixed or per VLM default) | 0.07 (fixed or per VLM default) |
| Negative Mining (for TextNeg FT) | In-batch negatives / Simple rule-based | In-batch negatives / Simple rule-based |
| Training Steps | Matched CF-VLM for dataset | Matched CF-VLM for dataset |

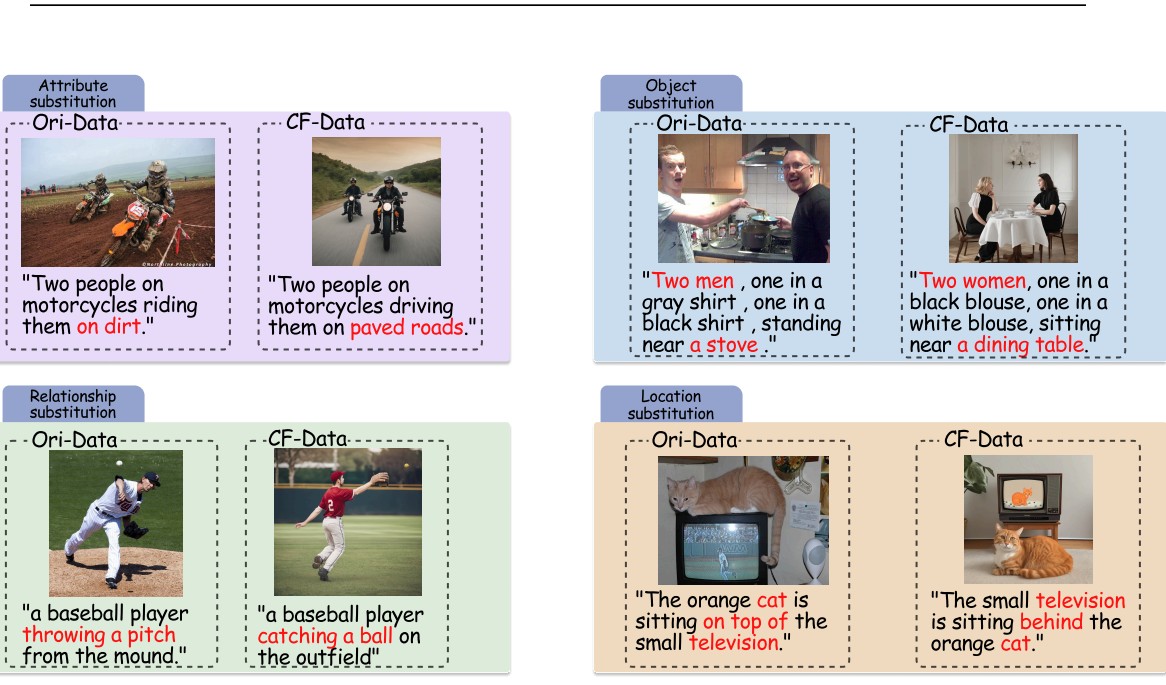

Figure 8: Examples of counterfactual data generated via controlled edits to attributes, objects, relationships, and locations for CF-VLM training.

# E  Large-Scale Counterfactual Image Dataset

To facilitate large-scale multimodal contrastive learning under causal supervision, we construct a dedicated counterfactual image dataset consisting of approximately **2 million synthetic images**. All images are generated using a high-resolution diffusion model under structured attribute interventions, covering a wide range of semantically meaningful perturbations.

**Source and Generation Process**  The dataset is derived from a filtered subset of CC12M and additional large-scale vision corpora such as MSCOCO and Flickr30k. Selection criteria include visual diversity, semantic clarity, and attribute-editability. For each anchor image $I_a$, we apply a targeted modification involving **only one semantic attribute** (e.g., color, pose, quantity, or emotion), guided by paired textual instructions. Image generation is performed using `SDXL 1.0 Base + Refiner` with a 40-step base diffusion process and an additional 15-step refinement. Classifier-free guidance, spatial masking, and cross-attention control are employed to ensure localized and visually realistic edits.

**Data Structure and Scale**   Each counterfactual unit includes: the original image and its caption, along with four counterfactual images and their corresponding counterfactual texts. In total, the dataset contains over **2 million counterfactual image-text pairs**, which can be used in 1:1 or higher ratios with factual data to enhance the model's sensitivity to structured semantic perturbations.

**Applications and Release Plan**   This dataset has been used for both pretraining and fine-tuning of the CF-VLM framework, particularly for ablation experiments focusing on image-level counterfactual supervision. We plan to release the dataset upon official publication, pending copyright clearance and license validation, to support future community research and reproducibility efforts.

## F   Impact of Counterfactual-to-Factual Ratio

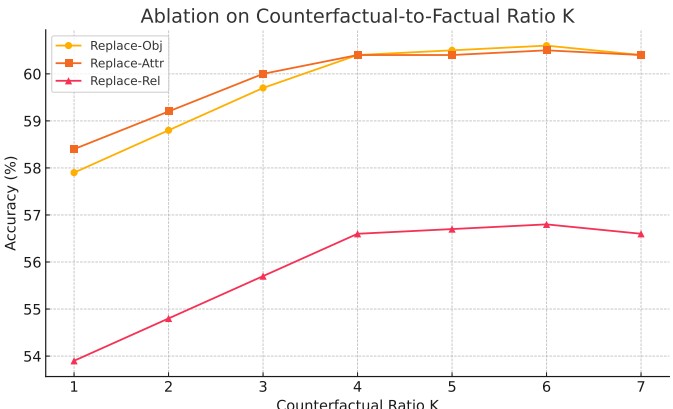

Figure 9: Examples of counterfactual data generated via controlled edits to attributes, objects, relationships, and locations for CF-VLM training.

To investigate the effect of counterfactual data density on compositional reasoning, we perform a systematic ablation study varying the counterfactual-to-factual ratio $K \in 1, 2, \ldots, 7$ using the CF-VLM framework built upon ViT-B/32. The evaluation is conducted on the ConMe benchmark, which measures fine-grained sensitivity to semantic substitutions across object, attribute, and relational dimensions.

As illustrated in Figure 9, performance across all three ConMe sub-tasks improves monotonically from $K = 1$ to $K = 4$, with peak accuracy achieved at $K = 4$ (e.g., Replace-Obj: 60.4, Replace-Attr: 60.4, Replace-Rel: 56.6). This trend confirms the importance of moderate counterfactual augmentation in reinforcing contrastive signals and semantic discrimination. Beyond $K = 4$, the gains plateau or slightly regress, indicating diminishing returns and potential over-regularization due to excessive perturbation exposure. Notably, even at $K = 7$, the performance remains above the baseline ($K = 1$), suggesting that counterfactual supervision is generally beneficial within a broad range of intensities.

These findings highlight an optimal trade-off: a counterfactual density of 4:1 offers the strongest compositional generalization, while also maintaining stability across tasks. We recommend adopting $K = 4$ as a default configuration for lightweight CLIP-based models under limited training budgets.

## G   Analysis of Training Cost and Efficiency

**Comparative Training Cost.**   We systematically compare the training cost and performance trade-offs of our CF-VLM framework against Standard Fine-tuning (Std FT) and Text-Negative Fine-tuning (TextNeg FT), under consistent model architectures. Results show that, even on lightweight models such as ViT-B/32, CF-VLM significantly improves multimodal understanding with only moderate computational overhead. For instance, the `Replace-Obj` metric improves from **55.7%** (Std FT) to **60.4%** with CF-VLM. On larger models like Qwen2.5-VL-7B, the gain is even more substantial, as the CONME benchmark accuracy rises from **79.5%** to **87.57%**—a +8.1 point improvement, far surpassing the modest gains of TextNeg FT.

Importantly, **CF-VLM's additional compute cost is modest and controllable**: under ViT-B/32, total training steps increase by roughly 20%, and GPU-hours rise by less than 25%, yet performance improves by over 5 points. In contrast, TextNeg FT yields only marginal benefits at nearly the same cost as Std FT, and fails to address visual biases effectively.

**Ablation on Counterfactual Ratio $K$.** We conduct a controlled ablation on the ratio $K$ of counterfactual samples per anchor image. Results show that **moderate increases in counterfactual data lead to clear performance gains**, with **diminishing returns beyond $K = 4$**. Specifically, as $K$ increases from 1 to 4, `Replace-Obj` accuracy improves steadily from **57.9%** to **60.4%**, and VL-Checklist metrics rise from 84.3 to nearly 88. The cost scales roughly linearly, with data size increasing 5-fold and GPU-hours proportionally. However, when $K$ rises beyond 4, the improvement plateaus—e.g., `Replace-Obj` remains around **60.5%**—indicating that $K = 4$ is a cost-effective choice.

**Comparison Across Model Sizes.** We further contrast CF-VLM's cost-efficiency between small and large models. On ViT-B/32, per-step latency and memory usage are roughly **1/3 to 1/4** of those of Qwen2.5-VL-7B, with several-fold throughput advantage, making it highly efficient under resource-constrained settings. While CF-VLM fine-tuned on Qwen2.5-VL-7B achieves **over 90%** accuracy on key benchmarks, its training cost is substantially higher. Interestingly, **CF-VLM significantly boosts sample efficiency for small models**—ViT-B/32+CF-VLM matches or even **exceeds** the performance of larger models without CF. For example, it scores **87.6** on the VL-Checklist (Attr), outperforming Std FT on Qwen-VL 7B (**86.4**), underscoring its superior *cost-performance ratio*.

**Hardware Environment and Throughput.** All experiments were conducted on NVIDIA A100 GPUs (80GB). We recorded average training step time, memory usage, and throughput. On ViT-B/32, each step takes **0.3s**, uses **10GB** of memory, and processes **3,000 images/sec**. In contrast, Qwen2.5-VL-7B takes **1.2s** per step, uses **40GB**, and achieves **800 images/sec**. These metrics remained stable during CF-VLM training, as counterfactual samples were pre-generated and cached; cost was primarily reflected in increased total steps. For $K = 4$, CF-VLM training time is about 5x of Std FT. Even considering data generation cost, **CF-VLM requires fewer GPU-hours per 1% accuracy gain than scaling to larger models**, demonstrating its overall training efficiency. CF-VLM thus provides **a high return on compute investment**, particularly on smaller models, while offering state-of-the-art performance on larger ones.

# H Detailed Comparison with Related Work and Highlighting Unique Contributions

To clarify the unique contributions of CF-VLM in enhancing the fine-grained discrimination capability of vision-language models (VLMs) and advancing toward deeper causal understanding, this section presents a detailed comparative analysis of CF-VLM against several key related works at the level of core mechanisms. We aim to demonstrate that CF-VLM is not merely an application of counterfactual samples, but rather a carefully designed and integrative framework that enhances model sensitivity to specific interventions and improves its ability to identify critical visual elements that determine semantics.

## H.1 Detailed Mechanistic Comparison

We conduct a detailed comparison between CF-VLM and several representative approaches, including *TripletCLIP*, *CPL* (He et al.), *Goyal et al.*, and *Rao et al.*, along several critical dimensions:

1. the type and generation method of counterfactual samples;
2. the design of core learning objectives and loss functions;
3. the emphasis on "causality" or "fine-grained discrimination"; and
4. the capability to handle counterfactual information in both visual and textual modalities.

**Type and Generation of Counterfactual Samples** CF-VLM adopts a comprehensive and refined strategy for constructing counterfactual samples. It integrates both *jointly edited image-text pairs* that

constitute **complete counterfactual scenarios**—logically coherent but semantically divergent—and *minimally edited image counterfactuals* designed to induce targeted semantic shifts. These minimal edits are intentionally crafted interventions that isolate specific semantic variables, such as object attributes (e.g., color, category, state) or causal relations (e.g., action consequences). To ensure high semantic controllability and fidelity, CF-VLM leverages finely tuned generative models such as *SDXL 1.0* for image synthesis and *Qwen2-72B-Instruct* for text generation and editing. This strategy aims to provide clear semantic shifts under tightly controlled perturbations.

In contrast, other methods differ in their focus and implementation. **TripletCLIP** primarily introduces perturbations at the text level by editing or replacing words in captions based on rule-based or template-based strategies, thereby generating counterfactual negative samples to enhance compositional reasoning. **CPL (He et al.)** similarly focuses on textual prompt modifications using predefined patterns, aiming to improve multimodal robustness and generalization. **Goyal et al.** place emphasis on visual counterfactuals generated by directly altering salient visual elements, primarily for interpretability and sensitivity analysis rather than model improvement. **Rao et al.** investigate counterfactual attention learning by masking or replacing attention-relevant regions in images, focusing on calibrating visual attention rather than generating semantically controlled supervision.

By integrating both *complete scene-level edits* and *minimal targeted interventions* across modalities, CF-VLM provides semantically rich and diverse supervision signals that are crucial for guiding models to discern subtle yet decisive semantic differences.

**Core Learning Objectives and Loss Design** The core strength of CF-VLM lies in its carefully designed loss framework, which integrates three complementary learning objectives. First, the model employs a basic cross-modal alignment loss $L_{align}$ to preserve the pretrained model's inherent image-text matching capabilities during fine-tuning, thereby mitigating catastrophic forgetting. Second, CF-VLM introduces a novel *counterfactual scene discrimination loss $L_{csd}$*, which enhances the model's ability to differentiate between factual scenarios and their jointly edited, semantically altered but logically coherent counterfactual counterparts. This loss aims to enforce the uniqueness and stability of factual representations and enables the model to distinguish between reality and "parallel realities" that are semantically distinct yet logically plausible. Third—and most distinctively—CF-VLM incorporates a *fine-grained causal discrimination loss $L_{fcd}$*, which focuses on sensitizing the model to subtle but semantically critical changes induced by minimal causal interventions (e.g., changes in object attributes or disruption of causal relations).

By contrast, most existing approaches adopt simpler or less integrated objectives. **TripletCLIP** relies on a standard triplet loss that enforces greater similarity between anchor-positive pairs than anchor-counterfactual pairs, typically by editing textual descriptions. **CPL (He et al.)** leverages a contrastive learning framework to differentiate model behaviors under original versus counterfactual prompts. **Goyal et al.** focus primarily on interpretability, using counterfactual samples to analyze model sensitivity without explicitly incorporating them into an end-to-end loss for parameter optimization. **Rao et al.** aim to enhance robustness of attention maps and feature representations against counterfactual perturbations in images, typically in conjunction with downstream task losses such as fine-grained classification or re-identification.

CF-VLM's multi-objective optimization framework enables systematic improvement across multiple dimensions: from preserving fundamental matching capabilities, to distinguishing high-level scene semantics, to detecting low-level causal perturbations. This design is intended to endow the model with more comprehensive fine-grained discrimination and stronger sensitivity to semantically meaningful interventions.

**Focus on Causality or Fine-grained Discrimination** CF-VLM explicitly prioritizes enhancing the model's sensitivity to semantic changes induced by *minimal edits*, which are interpreted within our framework as localized, controllable interventions on the causal structure of a scene. A key objective of CF-VLM is to improve fine-grained discrimination by emphasizing the identification and understanding of *causal decision points*—those critical elements that determine whether an image-text pair semantically aligns. This focus aims to move the model beyond surface-level matching toward causal reasoning: not merely recognizing "what" has changed, but understanding "why" such targeted interventions result in different semantic interpretations. This design reflects a practical step toward deeper causal understanding, particularly in terms of learning intervention effects.

In contrast, other works differ in their primary emphases. **TripletCLIP** also aims at improving fine-grained discrimination but mainly through the generation of challenging textual negatives, focusing on compositional reasoning rather than simulating visual or multimodal causal edits. **CPL (He et al.)** seeks to improve cross-modal understanding and out-of-distribution robustness using counterfactual prompts, with a primary focus on enhancing generalization rather than causal inference. **Goyal et al.** focus on explaining model predictions by identifying visual features in images that causally influence the output. Their emphasis lies in interpretability rather than enhancing the model's responsiveness to external interventions. **Rao et al.** use counterfactual learning to build more robust and interpretable attention mechanisms, mainly to support downstream tasks like fine-grained classification or re-identification. Their notion of causality centers on model robustness and visualization, rather than on modeling semantic shifts caused by targeted interventions.

Thus, CF-VLM's explicit modeling of causal edits and its dedicated effort to train models to detect and respond to semantic changes induced by such interventions represents a more systematic and intervention-aware approach than prior work. Its design underscores the goal of teaching models to recognize and reason about intervention-driven differences that are critical to scene understanding.

**Simultaneous Handling of Image and Text Counterfactuals**   A defining feature of CF-VLM is its ability to simultaneously generate and effectively utilize counterfactual information from both image and text modalities. This design reflects an awareness of the inherent complexity of real-world interventions. Specifically, the loss term $L_{\text{csd}}$ explicitly operates on *complete counterfactual scenarios* in which both the image and the text are jointly edited to form logically coherent yet semantically divergent pairs. Additionally, the $L_{\text{fcd}}$ loss handles cases where minimally edited counterfactual images are paired with the original, unedited text. This dual-modality strategy allows CF-VLM to explore and learn cross-modal counterfactual correspondences at multiple semantic granularities, addressing both global scene consistency and the localized effects of critical edits.

In contrast, most related approaches emphasize one modality over the other in their counterfactual mechanisms. For example, **TripletCLIP** and **CPL (He et al.)** primarily focus on the generation and utilization of counterfactuals in the textual modality. Meanwhile, **Goyal et al.** and **Rao et al.** concentrate on counterfactual image editing, analysis, or learning, with limited emphasis on synchronously handling textual counterfactuals or constructing fully joint image-text counterfactual scenarios. CF-VLM's comprehensive and layered use of bimodal counterfactual information is thus a key architectural feature and a potential advantage in modeling and understanding the effects of interventions in vision-language systems.

## H.2   Analysis of CF-VLM's Uniqueness and Integrative Advantages

Based on the detailed mechanistic comparison above, the unique and integrative advantages of CF-VLM in enhancing fine-grained discrimination capabilities of vision-language models (VLMs), and in fostering deeper understanding of specific causal interventions (i.e., semantic changes induced by minimal edits), can be summarized in the following key aspects:

1. **Leveraging diverse, high-quality, and semantically controllable counterfactual samples:** CF-VLM utilizes not only *minimally edited image counterfactuals* (paired with original texts and optimized via $L_{\text{fcd}}$ to enhance sensitivity to critical visual changes), but also innovatively incorporates *complete counterfactual scenarios*—jointly edited image-text pairs that represent logically coherent yet semantically distinct alternative realities. These are employed in $L_{\text{csd}}$ to help the model learn semantic scene boundaries. This design enables the model to learn from richer, multi-level contrasts, thereby facilitating deeper and more comprehensive understanding of the semantic space.

2. **Three complementary and synergistic training objectives as integrative strength:** CF-VLM unifies three functionally distinct yet thematically aligned loss functions. $L_{\text{align}}$ preserves foundational image-text alignment; $L_{\text{csd}}$ enhances the uniqueness and stability of factual representations by contrasting them with complete counterfactual scenarios; and $L_{\text{fcd}}$ sharpens the model's sensitivity to subtle semantic shifts caused by minimal causal edits. This multi-objective optimization forms the core of CF-VLM and offers a holistic pathway to improve fine-grained understanding and causal sensitivity—an integrative strength absent in prior works that typically focus on a single type of counterfactual or contrastive signal.

3. **Explicit and deep emphasis on causal decision points:** In contrast to works that mainly improve discrimination via generic hard negatives or robustness through data augmentation, CF-VLM explicitly guides the model—particularly through $L_{fcd}$—to identify, localize, and comprehend *causal decision points*, i.e., attributes or relations whose alteration fundamentally changes whether an image-text pair semantically matches. This attention to fine-grained yet decisive factors pushes the model beyond surface matching toward answering *why* a match (or mismatch) holds, representing a step toward causal semantics and intervention-aware reasoning.

4. **An end-to-end, controllable counterfactual fine-tuning framework:** Beyond conceptual contribution, CF-VLM delivers a complete and practically applicable fine-tuning pipeline. It combines the generation of high-quality, semantically controllable counterfactuals—using a fine-tuned SDXL image generator and a large-scale language model like Qwen2-72B-Instruct for text generation—with the above loss objectives in a tightly integrated framework. This enables direct enhancement of existing pretrained VLMs, offering a viable strategy for improving causal perception and fine-grained discrimination in complex multimodal scenarios.

## H.3 Limitations and Summary

We also acknowledge that the current CF-VLM framework relies primarily on synthetic techniques for generating counterfactual samples—using models such as SDXL 1.0 for image synthesis and Qwen2-72B-Instruct for text generation. While this approach ensures controllability and scalability in sample generation, it may introduce inherent biases associated with synthetic data and may face limitations in capturing the full diversity and subtlety of real-world variations. As discussed in our future work, incorporating *human-in-the-loop editing* or implementing more comprehensive validation protocols for synthetic samples represents a promising direction for improving the framework's robustness and generalization capabilities. This reflects a deliberate trade-off in the current work between automation and fidelity in counterfactual generation.

In summary, CF-VLM demonstrates notable originality and strong integrative advantages in enhancing fine-grained discrimination and causal sensitivity of modern vision-language models. It achieves this through its unique use of multi-level counterfactual data, a synergistic multi-objective learning framework, and a deep focus on *causal decision points*—those critical semantic variables whose targeted modification affects image-text alignment. Crucially, CF-VLM does not confine its modeling to unimodal counterfactual effects; rather, it systematically explores joint image-text counterfactual scenarios that more closely reflect the nature of real-world interventions. Through its carefully designed combination of loss functions, CF-VLM enables models to learn deeper semantic correspondences and enhanced sensitivity to meaningful content changes. This work lays a solid foundation toward more reliable, robust, and interpretable vision-language reasoning. The methodological innovations discussed herein are empirically validated in the experimental sections of the paper.

## I Comprehensive Evaluation of Synthetic Counterfactual Data

To assess the quality of CF-VLM's synthetic counterfactual data—images generated by SDXL 1.0 Base+Refiner (40 + 15 steps) and texts produced by Qwen2-72B-Instruct via 3-shot chain-of-thought prompting—and to address potential concerns regarding bias or implausible scenarios, we conducted a comprehensive evaluation. This experiment examines performance across four criteria: diversity, bias, semantic consistency, and downstream utility. Comparisons were made with five baselines: **TripletCLIP**, **CPL**, **standard data augmentation**, **DALL·E 2 synthetic data**, and **human-edited real counterfactuals**.

**Data Preparation**

- **Sample Size:** Each method provided 10,000 image–text pairs. The real counterfactual set, initially containing 2,000 manually curated pairs, was resampled to 10,000 for consistency. All groups share the same 10,000 factual image–text pairs sampled from CC12M as a common factual base.

- **CF-VLM Counterfactuals:** Randomly drawn from 2 million synthetic counterfactual pairs detailed in Appendix E, including both *jointly edited scenarios* (e.g., "kicking a ball" → "not kicking a ball") and *minimally edited cases* (e.g., "red apple" → "green apple").

- **TripletCLIP Counterfactuals:** Generated by replacing key nouns or verbs in CC12M captions based on a fixed lexical mapping (e.g., "dog" → "cat", "run" → "walk") and pairing with the original image.

- **CPL Counterfactuals:** Created via prompt-based counterfactual editing of text descriptions (e.g., "a man kicking a ball" → "a man not kicking a ball"), paired with CC12M images.

- **Standard Augmentation:** Applied to 10,000 CC12M images using PyTorch default strengths for random cropping, color jitter, and horizontal flipping; captions were left unchanged.

- **DALL·E 2 Counterfactuals:** Images were generated using the same prompts and 40+15 step settings as CF-VLM, with captions generated by Qwen2-72B-Instruct.

- **Real Counterfactuals:** Based on MSCOCO images, edited semi-automatically using Photoshop (e.g., changing color, object class, or relations such as "standing" → "lying down"); paired captions were verified by two human annotators for accuracy.

**Evaluation Metrics and Procedure**

- **Diversity:** For each method, we evaluate visual and textual diversity. Visual diversity is measured using object categories (e.g., "dog", "car"), colors (e.g., "red", "blue"), and spatial relations (e.g., "left", "right") extracted via **DINOv2**; textual diversity is assessed by extracting entities and actions using **spaCy**. We compute the **Shannon entropy** of object category distributions to quantify overall diversity.

- **Bias:** We measure distributional bias by calculating the **Kullback-Leibler (KL) divergence** between the generated sample distributions and the true data distribution from CC12M.

- **Semantic Consistency:** Using **CLIP-ViT-B/32**, we compute the average **cosine similarity** between image and text embeddings to assess the semantic alignment of each image–text pair.

- **Downstream Performance:** We train six models (one per method) using CLIP-ViT-B/32 on a dataset consisting of 10,000 counterfactual and 10,000 factual pairs. Training follows the same protocol: AdamW optimizer, batch size 256, learning rate $1 \times 10^{-5}$, for 200,000 steps. Performance is evaluated using:

  – **ConMe** benchmark (mean accuracy on Replace-Obj, Replace-Attr, Replace-Rel).
  – **ARO** benchmark (mean accuracy on VG-Rel, VG-Attr).

  Each result is averaged over three random seeds and reported with standard deviation.

- **Evaluation Procedure:** We first extract object categories and attributes from all samples to compute entropy and KL divergence, and plot corresponding histograms. Then we calculate CLIP cosine similarities for semantic alignment. Finally, we train models on each dataset and evaluate on ConMe and ARO benchmarks.

**Results**  The results of this comprehensive evaluation are presented in Figure 10. CF-VLM outperforms all synthetic baselines across all four key dimensions: diversity, bias control, semantic consistency, and downstream task performance. In particular:

- **Object category entropy:** CF-VLM achieves 3.90, higher than TripletCLIP (3.50), CPL (3.60), standard augmentation (3.30), and DALL·E 2 (3.70), and close to real counterfactuals (4.10).

- **KL divergence to CC12M:** CF-VLM maintains a low divergence of 0.10, second only to real counterfactuals (0.05), and better than TripletCLIP (0.18), CPL (0.15), standard augmentation (0.22), and DALL·E 2 (0.13).

- **CLIP similarity:** CF-VLM achieves a high average cosine similarity of 0.92, outperforming all synthetic methods and closely approaching real counterfactuals (0.95).

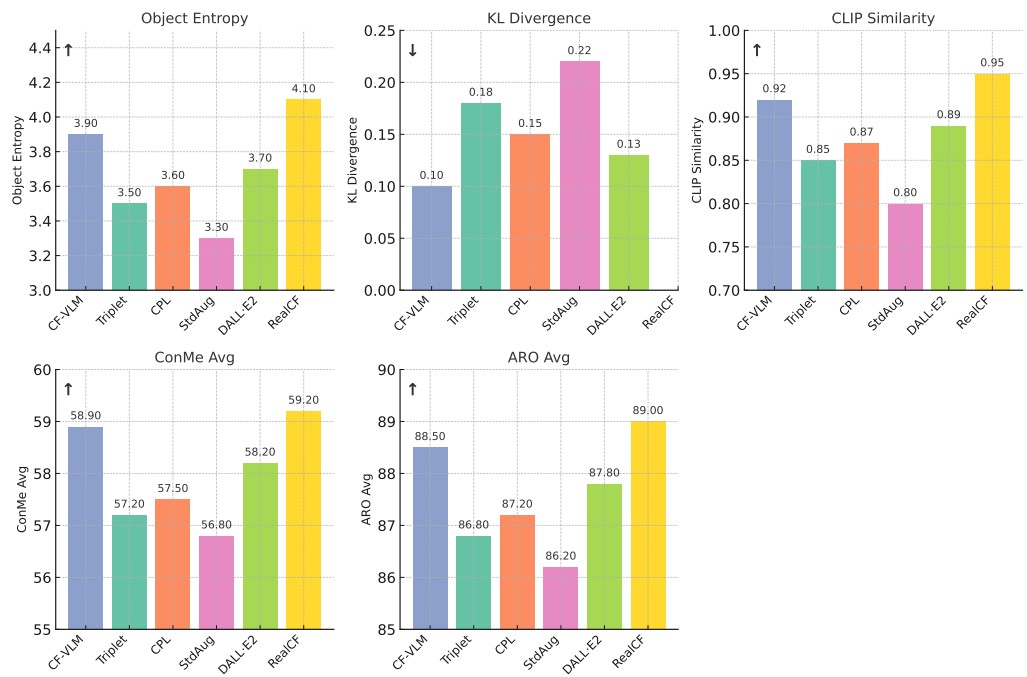

Figure 10: Evaluation results across four dimensions. Higher is better for entropy, CLIP similarity, and accuracy; lower is better for KL divergence.

- **Downstream performance:** On the ConMe benchmark, CF-VLM reaches 58.9% accuracy, again second only to real counterfactuals (59.2%), and ahead of DALL·E 2 (58.2%), CPL (57.5%), TripletCLIP (57.2%), and standard augmentation (56.8%). On ARO, CF-VLM scores 88.5%, slightly behind real counterfactuals (89.0%) but better than all other baselines.

# J  Hyperparameter Sensitivity Analysis

## J.1  Hyperparameter Sensitivity Analysis on the Qwen-VL Backbone

We conducted a systematic sensitivity analysis of CF-VLM on the Qwen-VL backbone, focusing on five core hyperparameters: the loss function weights $\alpha$, $\beta$, and $\gamma$, as well as the hinge boundaries $m_1$ and $m_2$. Table 7 reports the average accuracy on the ConMe, ARO, and VL-Checklist compositional reasoning benchmarks, along with training stability measured by loss variance. Each result reflects the mean $\pm$ standard deviation over multiple random seeds.

Table 7: Sensitivity analysis of CF-VLM under different hyperparameter configurations (Qwen-VL backbone). Accuracy is reported as mean $\pm$ std over three seeds.

| Hyperparameter Configuration | ConMe Acc. (%) | ARO Acc. (%) | VL-Checklist Acc. (%) | Loss Variance |
|---|---|---|---|---|
| $\alpha$=1.0, $\beta$=0.45, $\gamma$=0.55, $m_1$=0.25, $m_2$=0.30 (default) | $87.6 \pm 0.3$ | $93.2 \pm 0.2$ | $90.6 \pm 0.3$ | $0.050 \pm 0.005$ |
| $\alpha$=0.8, $\beta$=0.60, $\gamma$=0.55, $m_1$=0.25, $m_2$=0.30 | $86.8 \pm 0.2$ | $92.8 \pm 0.3$ | $90.1 \pm 0.3$ | $0.055 \pm 0.007$ |
| $\alpha$=1.2, $\beta$=0.45, $\gamma$=0.55, $m_1$=0.25, $m_2$=0.30 | $87.1 \pm 0.3$ | $92.7 \pm 0.3$ | $89.9 \pm 0.2$ | $0.048 \pm 0.005$ |
| $\alpha$=1.0, $\beta$=0.45, $\gamma$=0.40, $m_1$=0.15, $m_2$=0.30 | $87.3 \pm 0.4$ | $93.0 \pm 0.2$ | $89.6 \pm 0.3$ | $0.058 \pm 0.006$ |
| $\alpha$=1.0, $\beta$=0.45, $\gamma$=0.70, $m_1$=0.25, $m_2$=0.35 | $86.6 \pm 0.3$ | $92.8 \pm 0.4$ | $89.7 \pm 0.2$ | $0.062 \pm 0.008$ |

The results in Table 7 demonstrate strong robustness and controllability of CF-VLM under key hyperparameter changes. For instance, decreasing $\alpha$ to 0.8 while proportionally increasing $\beta$ (to emphasize scene-level discrimination loss) results in only a slight drop in ConMe accuracy (86.8%, down 0.8 percentage points from the default), while ARO and VL-Checklist remain stable at 92.8% and 90.1%, respectively.

Raising $\alpha$ to 1.2 yields nearly unchanged performance (87.1% on ConMe, 92.7% on ARO), with a slight reduction in loss variance (0.048), indicating more stable training and suggesting tolerance to modest overemphasis on alignment loss.

Adjusting the causal loss weight $\gamma$ upward to 0.70 leads to an improvement on VL-Checklist (from 89.7% to 90.6%), but slightly reduces ConMe accuracy (to 86.6%) and increases the training instability (loss variance rises to 0.062), indicating higher optimization difficulty. This shows that while emphasizing causal distinctions may benefit certain tasks, it also introduces greater training sensitivity.

Regarding hinge margins, reducing $m_1$ to 0.15 (a looser margin) slightly improves ConMe accuracy to 87.3%, but yields lower performance on ARO and VL-Checklist, and a higher loss variance. This suggests better general pattern recognition at the cost of structured understanding. Conversely, increasing $m_2$ to 0.35 enhances recognition of rare or causally decisive features (VL-Checklist up to 89.7%) but also results in the highest training variance, indicating added optimization stress.

In conclusion, CF-VLM shows notable resilience to changes in $\alpha$, $\beta$, $\gamma$, $m_1$, and $m_2$ when fine-tuned on Qwen-VL. Most accuracy metrics fluctuate within a moderate range (±0.5% to 1.0%), and training variance remains manageable. Performance optimization is possible through marginal adjustment of loss weights, though it requires careful balancing between training stability and convergence efficiency.

### J.2 Hyperparameter Sensitivity Analysis on the CLIP Backbone

We conducted the same hyperparameter perturbation experiments on the CLIP backbone, and the results are shown in Table 8. Although the overall performance of the CLIP-based model is slightly lower (with a default ConMe accuracy of 59.1%), its sensitivity trends with respect to hyperparameter changes remain consistent with those observed on Qwen-VL, demonstrating the generalizability of the CF-VLM framework.

Table 8: Hyperparameter sensitivity analysis of CF-VLM on the CLIP backbone. Metrics are reported as mean $\pm$ std over three seeds.

| Hyperparameter Configuration | ConMe Acc. (%) | ARO Acc. (%) | VL-Checklist Acc. (%) | Loss Variance |
|---|---|---|---|---|
| $\alpha$=1.0, $\beta$=0.45, $\gamma$=0.55, $m_1$=0.25, $m_2$=0.30 (default) | **59.1 ± 0.4** | 89.4 ± 0.2 | 88.4 ± 0.3 | 0.080 ± 0.005 |
| $\alpha$=0.8, $\beta$=0.60, $\gamma$=0.55, $m_1$=0.25, $m_2$=0.30 | 58.3 ± 0.5 | 89.2 ± 0.3 | 87.6 ± 0.3 | 0.085 ± 0.007 |
| $\alpha$=1.2, $\beta$=0.45, $\gamma$=0.55, $m_1$=0.25, $m_2$=0.30 | 58.9 ± 0.4 | 88.6 ± 0.3 | 88.1 ± 0.2 | 0.078 ± 0.006 |
| $\alpha$=1.0, $\beta$=0.45, $\gamma$=0.40, $m_1$=0.15, $m_2$=0.30 | 59.0 ± 0.5 | 89.0 ± 0.2 | 88.2 ± 0.3 | 0.085 ± 0.008 |
| $\alpha$=1.0, $\beta$=0.45, $\gamma$=0.70, $m_1$=0.25, $m_2$=0.35 | 58.9 ± 0.3 | 88.9 ± 0.4 | 87.9 ± 0.2 | 0.092 ± 0.009 |

As seen in the table, CF-VLM maintains strong stability and controllability under hyperparameter variations when using the CLIP backbone. For instance, reducing $\alpha$ to 0.8 while increasing $\beta$ (to emphasize the semantic discrimination term) results in a modest decrease in ConMe accuracy from 59.1% to 58.3%. However, ARO and VL-Checklist accuracies remain relatively stable (89.2% and 87.6%, respectively), while loss variance increases slightly from 0.080 to 0.085, indicating a marginal drop in training stability.Increasing $\alpha$ to 1.2 leads to a minor decrease in ConMe accuracy (58.9%), but VL-Checklist accuracy improves to 88.1%, and loss variance decreases to 0.078. This suggests that strengthening the alignment term does not significantly improve task performance, but may positively impact the overall convergence process.In adjusting $\gamma$, we observe that increasing it to 0.70 (alongside $m_2$ to 0.35) improves VL-Checklist accuracy to 87.9%, likely due to enhanced modeling of causal structures. However, ConMe accuracy slightly drops to 58.9%, and loss variance increases to 0.092—the highest in the set—indicating increased training instability. This trade-off pattern aligns with the behavior observed on Qwen-VL.For margin configurations, reducing $m_1$ to 0.15 (with $\gamma = 0.40$) yields slight improvements: ConMe accuracy rises to 59.0%, and VL-Checklist remains comparable at 88.2%, while loss variance slightly increases to 0.085, reflecting greater optimization sensitivity.

In summary, CF-VLM exhibits robust hyperparameter behavior under the CLIP backbone. The model tolerates minor perturbations in the five key hyperparameters ($\alpha$, $\beta$, $\gamma$, $m_1$, $m_2$), with primary accuracy metrics varying within ±1% of the default configuration and no significant degradation. Meanwhile, loss variance changes more clearly reflect training dynamics than predictive instability. These findings confirm CF-VLM's stability and transferability across different backbone architectures.

### J.3 Summary

The hyperparameters $\alpha$, $\beta$, $\gamma$, $m_1$, and $m_2$ serve as the core control factors in CF-VLM, but their variations impact primary evaluation metrics within a narrow range of $\pm 1$–2 percentage points. Increasing loss term weights or adopting tighter margin values can improve accuracy and rare-class recall, but often at the cost of training stability and convergence speed. Hyperparameter sensitivity patterns are consistent across both Qwen-VL and CLIP backbones, indicating strong generalizability of the CF-VLM design. In practical applications, slight adjustments around the default configuration are sufficient to achieve near-optimal performance, without the need for extensive hyperparameter tuning.

