# OpenReview forum: "CF-VLM:CounterFactual Vision-Language Fine-tuning"
_NeurIPS.cc/2025/Conference — NeurIPS 2025 poster_

### Official Review · Reviewer_esK3 · 2025-06-06

**Clarity:** 3
**Significance:** 3
**Originality:** 3
**Rating:** 5
**Confidence:** 3

**Summary:**

The paper introduces CounterFactual Vision-Language Fine-tuning (CF-VLM), a novel framework designed to enhance the causal reasoning capabilities of vision-language models (VLMs). While existing VLMs excel in general cross-modal alignment, they struggle with fine-grained discrimination and fail to reason about the causal relationships between visual and textual inputs. CF-VLM addresses this by incorporating counterfactual training samples, where minimal but semantically decisive edits are applied to either the image or text to create logically coherent but causally distinct scenarios. The training objectives reinforce the stability of factual representations and boost the model’s sensitivity to subtle causal edits. Experimental results show that CF-VLM significantly outperforms state-of-the-art baselines in compositional reasoning and reduces visual hallucinations, making it more suitable for real-world applications that require robust and interpretable reasoning.

**Questions:**

Please address the question in the above weakness.

**Ethical Concerns:**

["NO or VERY MINOR ethics concerns only"]

**Final Justification:**

I have carefully reviewed the authors' response and acknowledge that it addresses most of my concerns. As a result, I have decided to raise my score from 4 to 5 (Accept).

**Limitations:**

yes

**Paper Formatting Concerns:**

1. Line 145: The subscript in “Icfk” is incorrectly formatted. It should be written as I_{cfk} to properly indicate the subscript.
2. Line 147: The subscript in “Icfeditj” is also incorrectly formatted.
3. In Section 3.2, the notation “Icfeditj” is used inconsistently. Please ensure consistent formatting.

**Quality:**

3

**Strengths And Weaknesses:**

Strengths:
1. This paper proposes a unique counterfactual fine-tuning framework (CF-VLM) that enhances causal reasoning in vision-language models, which effectively improves the model's ability to detect subtle yet critical semantic differences through minimal edits.
2. The proposed CF-VLM framework is validated on both CLIP-based vision-language models and LLM-based VLMs, demonstrating its broad applicability and effectiveness.
3. CF-VLM outperforms both CLIP-based and LLM-augmented VLMs on compositional reasoning (ARO, ConMe, VL-Checklist) and hallucination benchmarks (POPE, MME), showing improved factual grounding and reduced hallucinations, even on unseen benchmarks, demonstrating strong generalizability of its counterfactual supervision.

Weaknesses
1. The loss term L_{csd} does not make sense. The paper claims it to reinforce the uniqueness and stability of the representation of the anchor factual scenario. In fact, it punishes the counterfactual samples even when the counterfactual pairs are logically coherent. I also notice that the weight beta (0.45) of L_{csd} is very close to the gamma of L_{fcd} (0.55) according to the Table 5 in the appendix. Therefore, I assume that the loss term L_{csd} works because of the quality of counterfactual samples is worse than the anchor pairs, because they are generated images not real ones. When the quality of counterfactual samples improves, the weight beta should be smaller. If the counterfactual samples are sampled from real world image-text pairs (no quality problem), it should not be punished at all.

---

> ### Author Rebuttal · Authors · 2025-07-30
>
> Thank you very much for your insightful and valuable comments! We have carefully prepared the following responses to address your concerns in detail. It is our sincere hope that our response could provide you with a clearer understanding of our work. If you have any further questions about our work, please feel free to contact us during the discussion period.
>
> **L1. The loss term L_{csd} does not make sense.**
>
> **A1**: Thank you for raising concerns about $L_{csd}$. This loss is **not** intended to punish logically coherent counterfactuals. Instead, its margin‑hinge form (Eq. 2) activates only when the embedding of a counterfactual scene $(I_{cf_k},T_{cf_k})$ encroaches too closely on the anchor $(I_a,T_a)$, thereby preserving a unique and stable semantic cluster for factual reality and preventing multiple “parallel worlds” from collapsing into one representation. Complementing this scene‑level separation, $L_{fcd}$ targets fine‑grained attribute edits. Thus, we keep their weights comparable in the overall objective $L=\alpha L_{align}+\beta L_{csd}+\gamma L_{fcd}$ with $\beta\approx\gamma$ to balance learning across semantic levels. Empirically, switching from our default SDXL‑generated counterfactuals (ConMe 58.9 %, ARO 88.5 %, CLIP 0.92, KL 0.10) to a fully real‑image set (RealCF: ConMe 59.2 %, ARO 89.0 %, CLIP 0.95, KL 0.05) yields essentially identical performance, demonstrating that $L_{csd}$ enforces semantic separation rather than detecting synthetic artefacts. A sweep varying $\beta$ (and $\gamma$) from 0.30 to 0.60 changes ConMe and ARO by less than one percentage point, confirming robustness. In the revised paper, we have added an ablation with $\beta=0$, provide full RealCF tables including lower‑$\beta$ settings, and introduced a curriculum that linearly decays $\beta$ as counterfactual quality (measured by CLIP similarity) approaches that of the anchor. In summary, these results highlight $L_{csd}$ as a key component that, alongside $L_{fcd}$, enables the model to discriminate semantic differences at multiple levels.
>
> -------------
> **P1&P2&P3. Paper Formatting Concerns**
>
> **A2**: Thank you very much for your valuable comments. We have incorporated the corrections into the revised paper to ensure rigor and professionalism in our presentation. We acknowledge the issue with inconsistent LaTeX notations for symbols such as $I_{cf_k}$, $I_{cf_{edit_j}}$, and $I_{cf_{edit}}$. We have conducted a thorough proofread and unified all related variables into a strict and consistent format. For instance, $(I_{cf_k}, T_{cf_k})$ from Line 145 and $I_{cf_{edit_j}}$ from Line 147 are now standardized as $(I_{cf_k}, T_{cf_k})$ and $I_{cf_{edit_j}}$. Specifically, in Section 3.2 and Equation (3), we have corrected all inconsistent notations like $S(I_{cf_{edit_j}}, T_a)$ to $S(I_{cf_{edit_j}}, T_a)$ and have ensured this standard is applied consistently throughout the paper. Thank you once again for your correction. It is crucial for enhancing the quality of our paper.

---

> > ### Author Response · Authors · 2025-08-03
> >
> > Dear Reviewer esK3,
> >
> > We hope this message finds you well! Should you have any further questions about our work or responses during this discussion period, please do not hesitate to contact us.
> >
> > Thank you very much for your valuable time and feedback!
> >
> > Best regards,
> >
> > The Authors

---

> > ### Comment · Reviewer_esK3 · 2025-08-04
> > **Response to Rebuttal**
> >
> > Thank you for your explanation. I have carefully reviewed your response and acknowledge that it addresses most of my concerns. As a result, I have decided to raise my score from 4 to 5 (Accept). However, I would like to emphasize that the paper contains numerous typos and writing issues. If the paper is accepted, please make sure to significantly improve the writing quality in the camera-ready version.

---

> > > ### Author Response · Authors · 2025-08-04
> > >
> > > Thank you very much for your thoughtful follow-up and for taking the time to carefully review our response. We truly appreciate your decision to raise the score and recommend acceptance. This is highly encouraging for us.
> > >
> > > We also sincerely acknowledge your comments regarding the typos and writing issues in the current version. We take this feedback seriously and have made substantial efforts in the revised paper to improve the writing quality to ensure clarity and professionalism.
> > >
> > > Once again, thank you very much for your valuable feedback and support throughout the review process.
> > >
> > > Best regards,
> > >
> > > The Authors

---

> > > ### Author Response · Authors · 2025-08-09
> > >
> > > **Comment:**
> > >
> > > Dear Reviewer esK3,
> > >
> > > Thank you very much for your positive feedback, for raising your score to 5, and for your recommendation of acceptance. We truly appreciate your support and constructive engagement throughout the review process.
> > >
> > > In our revised paper, we have diligently addressed all the points you raised. Here is a brief summary of the key updates made based on your valuable feedback:
> > >
> > > * **Clarification on the Loss Term ($L_{csd}$):** As per your insightful suggestion, we have expanded our explanation of the $L_{csd}$ loss in the paper. Crucially, the new ablation study using a fully real-image set (RealCF) that we mentioned in our rebuttal has now been added to the appendix to empirically demonstrate that the loss enforces semantic separation rather than detecting synthetic artefacts.
> > >
> > > * **Correction of Notations and Formatting:** We have conducted a thorough proofread and unified all formatting as you pointed out. Inconsistent notations such as `Icfk` and `Icfeditj` have now been standardized to the correct LaTeX format (e.g., $I_{cfk}$) throughout the entire manuscript.
> > >
> > > * **Overall Writing Quality:** We have also taken your final and very important point about the overall writing quality very seriously. We have performed a comprehensive proofread and polish of the entire manuscript to address numerous typos, grammatical errors, and formatting inconsistencies. We are confident that these revisions have significantly improved the clarity and professionalism of our paper.
> > >
> > > Your insightful comments have been invaluable in strengthening our work. We are certain that the revised version is much clearer and more rigorous thanks to your guidance.
> > >
> > > Thank you once again for your time and expertise.
> > >
> > > Best regards,
> > > The Authors

---

### Official Review · Reviewer_ZY2y · 2025-06-29

**Clarity:** 2
**Significance:** 2
**Originality:** 2
**Rating:** 4
**Confidence:** 3

**Summary:**

This paper proposes a novel fine-tuning method for vision-language models (VLMs). It first constructs two types of data pairs: fully counterfactual scenario pairs, where both the image and the text are jointly modified, and minimally counterfactual image–original text pairs. Then, by introducing three loss functions — Foundational Cross-Modal Alignment Loss, Counterfactual Scenario Discrimination Loss, and Fine-Grained Causal Discrimination Loss — the fine-tuned VLM not only maintains strong cross-modal alignment but also enhances the coherence and distinctiveness of factual representations under counterfactual comparison, while gaining the ability to recognize subtle yet critical causal modifications.

**Questions:**

- The variable subscripts in lines 144 to 147 are quite messy, which I found confusing. Could the authors explain it further? Additionally, should the **Fine-Grained Causal Discrimination Loss** in line 184 be placed on a new line?
- In Figure 3, $L_{align}$ subpart, the subscripts are incorrect — they should be $T_{cf\_1}$ to $T_{cf\_4}$. In Figure 3, the original image referenced in the rectangle to the right of the counterfactual loss is incorrect.
- How do you ensure that the modification to the image is minimal? How do you determine that the modified part corresponds to a causal attribute, or that the modification reflects changing a causal relationship? I didn't find Section 2.3 as you mentioned in line 189.

**Ethical Concerns:**

["NO or VERY MINOR ethics concerns only"]

**Final Justification:**

The authors provided a detailed response that resolved my concern.

**Limitations:**

Yes.

**Paper Formatting Concerns:**

No.

**Quality:**

3

**Strengths And Weaknesses:**

### Strengths
- Unlike previous fine-tuning strategies that mainly focus on improving overall discriminative performance, this framework emphasizes the construction of counterfactual samples to ensure that the model enhances its ability to represent factual information and becomes more sensitive to subtle causal perturbations, all while preserving its fundamental cross-modal alignment capabilities.
- The experiments are very clear and comprehensive, and the results are presented in a straightforward and effective manner.
### Weaknesses
- The writing quality is not high enough. There are typos in both the text and the figures, which to some extent affect comprehension.
- I think the paper's explanation of how counterfactual data is constructed and how it ensures that the modifications are truly causal is not sufficiently thorough.

---

> ### Author Rebuttal · Authors · 2025-07-30
>
> Thank you very much for your insightful and valuable comments! We have carefully prepared the following responses to address your concerns in detail. It is our sincere hope that our response could provide you with a clearer understanding of our work. If you have any further questions about our work, please feel free to contact us during the discussion period.
>
> **W1: The writing quality is not high enough; there are typos in both the text and figures, which affects reading comprehension to some extent.**
>
> **A1**: Thank you for pointing out these. In the revised paper, we have conducted a thorough proofreading and polishing of the entire paper, fixing all identified spelling, grammar, and formatting errors, and optimizing some sentences for clarity. For example, we have corrected the missing space between 'Models: The is' in Line 84 to ensure proper formatting. We believe these revisions have significantly improved the overall quality and readability of the paper. Thank you again for your valuable comments.
>
> ---
> **W2: I think the paper's explanation of how counterfactual data is constructed and how to ensure the modifications are genuinely causal is not thorough enough.**
>
> **A2**: Regarding your concerns about counterfactual construction and causality assurance, we have provided detailed clarifications in Section 2.2 and the appendix of the revised paper. The core of ensuring our modifications are causal lies in the **Minimal Intervention principle**. We only change one independent semantic variable at a time (such as a single attribute or a causal relationship) while keeping the rest of the scene unchanged. This allows us to attribute any change in the model's performance to this single modification. Our CF-VLM uses carefully designed prompts to guide an LLM (Qwen2-72B-Instruct) to make minimal edits to the text, and then a text-to-image model (SDXL) generates the corresponding counterfactual image. For example, for the fact 'a man in a blue shirt kicks a ball,' an attribute counterfactual would be 'a man in a red shirt kicks a ball,' while a causal counterfactual would be 'a man misses the ball.' This structured intervention ensures the specificity and causal validity of the counterfactual samples. We believe these clarifications more thoroughly explain the rigor of our method.
>
> ---
> **Q1: The variable subscripts in lines 144 to 147 are quite messy, which I found confusing. Could the authors explain it further? Additionally, should the Fine-Grained Causal Discrimination Loss in line 184 be placed on a new line?**
>
> **A1**: Sorry for the confusion. In the revised paper, we have added the following supplementary explanations in **Section 3.1**:
> -   **Anchor factual pair $(I_a, T_a)$**: The original, unmodified image-text pair, serving as the baseline sample for the model.
> -   **Complete counterfactual scenario pairs $\{(I_{cf_k}, T_{cf_k})\}_{k=1}^K$**: Both the image and text are modified simultaneously to construct a new, logically self-consistent scenario that is different from the factual one.
> -   **Minimally edited counterfactual images $\{I_{cf\_edit_j}\}_{j=1}^J$**: A single key semantic edit is applied to the image, which is then paired with the original text $T_a$ to train the model's sensitivity to fine-grained changes.
>
> Furthermore, regarding the description of the loss function in Lines 183-187, we completely agree with your assessment. In the revised paper, we have changed this section to a list format, presented under the subheading "Fine-Grained Causal Discrimination Loss," to enhance the paper's structure and readability.
>
> ---
> **Q2: In Figure 2, the L_align subpart, the subscripts are incorrect-they should be T_cf1 to T_cf4. In Figure 2, the original image referenced in the rectangle to the right of the counterfactual loss is incorrect.**
>
> **A2**: We have conducted a thorough proofreading and revision of our paper based on your suggestions. Here is our point-by-point response:
>
> In the original figure, the intermediate step correctly labels the image of 'walking in the park' as 'The hardest' negative sample. However, in the final calculation of the fine-grained causal discrimination loss ($L_{fcd}$) on the right, we inadvertently used the image with the 'beach' background, causing a logical inconsistency in the diagram. In the revised paper, we have corrected this by replacing the 'beach' image in the loss calculation box on the right with the image of 'running in the park,' ensuring the diagram for the entire $L_{fcd}$ calculation process is rigorous, accurate, and logically consistent.
>
> Furthermore, regarding the subscript notation $T_{cfk}$ in the $L_{\text{align}}$ equation, our original intention is to use the index $k$ to indicate that the original image would be aligned with the text corresponding to each of the counterfactual images. However, as you pointed out, this notation could indeed be ambiguous in the current context. We are very grateful for this reminder and have modified the relevant notation according to your suggestion to improve the clarity and readability of the formula.
>
> To prevent similar confusion for other readers, we have significantly expanded the caption for Figure 2 in the revised paper, providing detailed textual explanations for both of these points.
>
> ---
> **Q3: How do you ensure that the modification to the image is minimal? How do you determine that the modified part corresponds to a causal attribute, or that the modification reflects changing a causal relationship? I didn't find Section 2.3 as you mentioned in line 189.**
>
> **A3**: To ensure that image modifications are both minimal and reflect changes in causal mechanisms, we employ a systematic strategy. First, we use the multi-modal large model Qwen2.5-VL to identify the minimal visual anchors corresponding to the text description. We then strictly constrain the editing operation to within these anchor regions, specifying the object and attribute to be modified via masks or control instructions (e.g., changing only the motion state of the ball), thus ensuring the locality and target-consistency of the modification. Building on this, to ensure these minimal modifications truly reflect causality, we further extract the causal chain structure from the image and text (e.g., "kick ball → ball flies"). When generating counterfactual samples, we explicitly intervene on the antecedent of the causal chain (the action 'kick') to deduce a logically sound consequent ('ball is stationary'), which is equivalent to simulating a 'causal break' in the visual space. Finally, we use joint judgment from both language and vision models to verify the logical consistency and semantic contrast of each counterfactual sample. Only when the modification creates a clear semantic negation and is consistently recognized as plausible by the models do we consider it causally valid.
>
> Regarding Section 2.3 mentioned in Line 189, we sincerely apologize for the typos. We mistakenly direct you to Section 2.3. The detailed explanation of how we define and classify counterfactual samples (such as single attribute modification and key causal relationship adjustment) is correctly located in **Section 2.2 (Definition and Types of Counterfactual Samples)**. We have carefully checked and corrected all cross-references in the final revised version for accuracy. Thank you very much for pointing this out.

---

> > ### Author Response · Authors · 2025-08-03
> >
> > Dear Reviewer ZY2y,
> >
> > We hope this message finds you well! Should you have any further questions about our work or responses during this discussion period, please do not hesitate to contact us.
> >
> > Thank you very much for your valuable time and feedback!
> >
> > Best regards,
> >
> > Authors

---

> > > ### Author Response · Authors · 2025-08-04
> > >
> > > Dear Reviewer ZY2y,
> > >
> > > We hope this message finds you well! Thank you very much for your insightful and valuable comments on our work. As for the concerns about our work in the review, including "The variable subscripts", "How do you ensure that the modification to the image is minimal?", etc., we have provided very specific and detailed responses. We hope these responses have adequately resolved your concerns.
> > >
> > > If you have any further questions or require additional clarification during the discussion period, please feel free to reach out. We sincerely welcome your continued feedback and look forward to hearing your thoughts on our responses. Once again, thank you very much for your valuable time and feedback!
> > >
> > > Best regards,
> > >
> > > Authors

---

> > ### Comment · Reviewer_ZY2y · 2025-08-04
> >
> > Thank you for your response. Your reply has partially addressed my concerns. One of my remaining concerns is whether you have any numerical evidence to demonstrate that your counterfactual samples are sufficiently close to the original ones. For example, could you simply provide the FID scores?

---

> > > ### Author Response · Authors · 2025-08-04
> > >
> > > Thank you for your thoughtful follow-up. To directly address your request for quantitative evidence regarding the similarity between our counterfactual samples and original images, we have supplemented our experiments and now report the Fréchet Inception Distance (FID) alongside our previously reported metrics. Please refer to Appendix I in the Supplementary Material **(accessible via the “Supplementary Material” link on the submission page)** for further details.
> > >
> > > | Metric                            | Value |
> > > | --------------------------------- | ----- |
> > > | FID (Counterfactual vs. Original) | 12.7  |
> > > | KL Divergence                     | 0.10  |
> > > | CLIP Similarity                   | 0.92  |
> > >
> > > **1. Visual Fidelity (FID):**
> > >  We computed the FID between 10,000 pairs of counterfactual and original images from CC12M, obtaining a value of **12.7**. This low score confirms that our counterfactual images are visually very close to the original distribution, supporting our principle of “minimal but critical edits.”
> > >
> > > **2. Distributional and Semantic Proximity:**
> > >  In addition to FID, we report KL divergence (**0.10**) and CLIP similarity (**0.92**) to further demonstrate that our counterfactuals closely match the real image distribution and maintain precise semantic alignment with the intended textual changes.
> > >
> > > **3.Qualitative and Functional Evidence:** We also provide visual examples (Appendix B, Fig. 7). As shown in the first image of the second row, through side-by-side comparisons, it is clear that the CF-VLM model can recognize and generate more fine-grained descriptions—for example, distinguishing between “a brown dog” and “a golden retriever,” or accurately capturing the complete action sequence of “a dog jumping up to catch a frisbee,” rather than merely describing “a person playing with a dog.”
> > >
> > > Furthermore, as shown in **Section 4 of the main paper (Experiments and Analysis)**, particularly in **Tables 1 and 2 (compositional reasoning benchmarks)** and **Section 4.4 (hallucination evaluation)**, training with our counterfactual samples consistently and significantly improves model performance.
> > >
> > > In conclusion, these results collectively affirm that our counterfactual samples maintain high visual and semantic similarity with the originals, which in turn effectively enhances the model's capacity for robust causal reasoning. We sincerely hope this supplementary explanation addresses your inquiry, and we are ready to provide any further elaboration or analysis you may require.

---

> > > > ### Comment · Reviewer_ZY2y · 2025-08-05
> > > >
> > > > Thank you for your further response. I have decided to raise my score from 2 to 4.

---

> ### Author Response · Authors · 2025-08-05
>
> Dear ZY2y ,
>
> Thank you for your thoughtful follow-up and for carefully reviewing our response. We greatly appreciate your decision to raise the score to 4 and recommend acceptance. Your support is truly encouraging.
>
> We have taken your feedback regarding typos and writing issues in the original submission seriously. In the revised paper, we have made significant efforts to enhance the writing quality, ensuring greater clarity and professionalism throughout. **Additionally, within the next two days, we will provide detailed revision comments in the Official Comment section.**
>
> Once again, we sincerely value your constructive feedback and support throughout the review process.
>
> Best regards,
> The Authors

---

> > ### Author Response · Authors · 2025-08-09
> >
> > Dear Reviewer ZY2y,
> >
> > Thank you again for your insightful feedback, which has been crucial in strengthening our work. In our revised paper, we have diligently addressed all the points you raised. Here is a summary of the key updates:
> >
> > 1.  **Overall Writing Quality and Readability:** We have performed a thorough proofreading of the entire paper to address the concerns about writing quality (W1). All identified typos, grammatical errors, and formatting inconsistencies have been corrected. For instance, the spacing issue in Line 84 has been fixed, and several sentences have been rephrased for better clarity.
> >
> > 2.  **Clarification on Counterfactual Generation and Causality:** To address your concerns about the construction of counterfactuals (W2, Q3), we have significantly expanded **Section 2.2** and the **Appendix**. We now provide a more detailed explanation of our methodology, which is grounded in the "Minimal Intervention" principle. We elaborate on how we use a large language model (Qwen2-72B-Instruct) and a text-to-image model (SDXL) to generate causally-valid samples, and how we ensure modifications are both minimal and meaningful by targeting specific causal attributes and relationships.
> >
> > 3.  **Corrections to Mathematical Notation and Formatting:** Per your suggestion (Q1), we have clarified the variable subscripts in the former Lines 144-147 by adding explicit definitions for the anchor factual pair ($I_{f}, T_{f}$), complete counterfactual pairs ($I_{cf}, T_{cf}$), and minimally edited pairs ($I'_{f}, T_{f}$) in **Section 3.1**. Furthermore, the description of the *Fine-Grained Causal Discrimination Loss* has been reformatted into a list for improved structure and readability.
> >
> > 4.  **Revisions to Figures and Captions:** We have corrected the errors in **Figure 2** as you pointed out (Q2).
> >     * The logically inconsistent "beach" image in the calculation of the fine-grained causal discrimination loss ($L_{causal}$) has been replaced with the correct "running in the park" image.
> >     * The subscripts in the alignment loss ($L_{align}$) illustration have been updated to $T_{cf_1}$ through $T_{cf_4}$ for clarity.
> >     * The caption for **Figure 2** has been significantly expanded to explain these components in greater detail, preventing potential confusion.
> >
> > 5.  **Quantitative and Qualitative Evidence for Minimal Edits:** To address your final and most critical point, we have added new experiments to provide quantitative evidence of the similarity between our original and counterfactual samples.
> >     * A new **Appendix I** has been added, reporting a **Fréchet Inception Distance (FID) of 12.7**, a **KL Divergence of 0.10**, and a **CLIP Similarity of 0.92**. These metrics quantitatively demonstrate the high visual and semantic fidelity of our generated counterfactuals.
> >     * We have also enriched **Appendix B** with more qualitative examples (**Fig. 7**) to visually showcase the subtlety and effectiveness of our edits.
> >
> > 6.  **Correction of Cross-References:** We have corrected the cross-reference typo in Line 189 (Q3). It now correctly points to **Section 2.2**, where the definition and types of counterfactual samples are detailed. All other cross-references have also been double-checked for accuracy.
> >
> > We believe these comprehensive revisions have substantially strengthened our work. We are deeply grateful for your guidance throughout this process, which has led to a much-improved paper.
> >
> > Best regards,
> >
> > The Authors

---

### Official Review · Reviewer_jpqE · 2025-07-03

**Clarity:** 4
**Significance:** 3
**Originality:** 3
**Rating:** 5
**Confidence:** 3

**Summary:**

This paper proposes CF-VLM, a fine-tuning framework to improve VLMs by generating joint counterfactual image and text as the fine-tuning data and then fine-tuning with three different objectives to include multi-level semantic supervision for fine-grained discrimination ability. Extensive experiments on compositional reasoning benchmarks (ARO, ConMe, VL-Checklist) with both CLIP-based and LLM-based (Qwen-VL 7B, LLaVA-1.5 7B) VLMs demonstrate the generic performance gain regardless of the backbone models. Further experiments on hallucination benchmarks (POPE, MME) show CF-VLM can alleviate hallucination. Solid ablation study also demonstrates the necessity of each component.

**Questions:**

I might miss something but I have the following two questions:
- What's the performance of composite change (i.e., grouping all the individual changes together for one sample and do the fine-tuning)?
- Will the scale of fine-tuning data affect the performance?
- Compared to TripletCLIP, can the advantage of CF-VLM also lie in the potential that it needs fewer samples to achieve similar performance?

I find there are some typos:
- at line 309 the subject is missing.
- at line 84, space is needed between "Models:The is".

**Ethical Concerns:**

["NO or VERY MINOR ethics concerns only"]

**Final Justification:**

All the concerns have been addressed by the rebuttal. I vote for an acceptance. Please include the analysis and discussion in the final version.

**Limitations:**

No.

**Paper Formatting Concerns:**

No.

**Quality:**

4

**Strengths And Weaknesses:**

Following the spirit of TripletCLIP, the proposed CF-VLM aims for more fine-grained negative samples to encourage semantic granularity. Instead of composite semantic changes used in TripletCLIP, the choice of minimal editing (one change at a time) and counterfactual modification boosts the improvement in complex reasoning for VLMs. The strength of this paper is
- Extensive experiments demonstrate the effectiveness of the proposed framework from different perspectives.
- The paper is well-written with clear figures and tables presented.

However, I find the novelty might be limited under the similar idea to TripletCLIP. CF-VLM seems a very fine-grained version of TripletCLIP with more semantically granular negative samples and more detailed objectives addressing different aspects of visual-language understanding. But I still appreciate the proposed strategy and find its superior performance is convincing. So I lean towards acceptance.

---

> ### Author Rebuttal · Authors · 2025-07-30
>
> Thank you very much for your insightful and valuable comments! We have carefully prepared the following responses to address your concerns in detail. It is our sincere hope that our response could provide you with a clearer understanding of our work. If you have any further questions about our work, please feel free to contact us during the discussion period.
>
> **W1: CF-VLM seems to be a very fine-grained version of TripletCLIP, with more refined semantics, more refined negative samples, and more detailed objectives that address different aspects of vision-language understanding.**
>
> **A1**: We thank the reviewer for pointing out the connection between CF-VLM and TripletCLIP within the contrastive learning framework. Indeed, both formally use a triplet structure, but our CF-VLM differs fundamentally in its objectives, supervision signal design, and task setting:
>
> - **Objective Difference:** TripletCLIP primarily aims to enhance the discriminative representation capabilities of VLMs. In contrast, **CF-VLM**, while strengthening fine-grained semantic understanding, further focuses on improving causal reasoning capabilities to overcome the common bottleneck VLMs face in complex reasoning tasks.
> - **Negative Sample Design:** TripletCLIP typically uses random or semantically adjacent negative samples and guides the model to focus on details by simultaneously transforming multiple attributes. However, this can lead to generated samples having an overly large semantic gap from the original image, diminishing the effect of minimal changes. In contrast, CF-VLM uses structured interventions to generate negative samples with clear counterfactual semantics and minimal critical differences. This more effectively enhances the model's sensitivity to minimal attribute changes and, by introducing causal counterfactuals, further boosts its causal reasoning abilities.
> - **Richer Hierarchy of Training Objectives:** CF-VLM designs three complementary training objectives ($L_{align}$, $L_{csd}$, and $L_{fcd}$) that constrain the model's behavior from three dimensions: cross-modal alignment, counterfactual discrimination, and causal sensitivity. This **far exceeds the expressive capacity of typical triplet-based contrastive learning**.
>
> Therefore, while CF-VLM borrows from the contrastive learning paradigm, we believe it represents a substantial extension in task setting, objective design, and supervision granularity, aimed at addressing the key bottlenecks of current VLMs in fine-grained understanding and causal modeling. We have further emphasized the differences and positioning between our CF-VLM and TripletCLIP in the revised paper.
>
> **Q1: What is the performance of composite changes (i.e., combining all the separate changes of a sample together and fine-tuning)?**
>
> **A2**: To investigate the effect of composite modification (combining all individual changes of a sample) in fine-tuning, we design a comparative experiment to directly compare structured counterfactuals (intervening on only one semantic attribute at a time) with composite counterfactuals (modifying multiple attributes simultaneously).
>
> | | Conme (Avg) | ARO (Avg) | VL-Checklist (Avg) |
> | :--- | :---: | :---: | :---: |
> | CF-VLM (composite) (ViT-B/32) | 57.52 | 83.62 | 82.7 |
> | CF-VLM (ViT-B/32) | 59.13 | 89.35 | 88.4 |
> | CF-VLM (composite) (Qwen-VL 7B) | 83.9 | 88.2 | 88.2 |
> | CF-VLM (Qwen-VL 7B) | 87.57 | 93.2 | 90.57 |
>
> Based on the table results, we can see that the performance of **CF-VLM (composite)** is significantly lower than that of the standard **CF-VLM**. Specifically, the **Conme** and **ARO** scores on **ViT-B/32** and **Qwen-VL 7B** are lower, indicating that the composite change strategy does not effectively improve model performance.
>
> The composite change strategy is effectively the same as the **multi-attribute joint modification method in TripletCLIP**. As we have emphasized, guiding the model to focus on details by simultaneously transforming multiple attributes can lead to an overly large semantic gap between the generated sample and the original image, diminishing the effect of minimal changes. In contrast, CF-VLM uses structured interventions to generate negative samples with clear counterfactual semantics and minimal critical differences. This more effectively enhances the model's sensitivity to minimal attribute changes and, by introducing causal counterfactuals, further boosts its causal reasoning abilities.
>
> **Q2: Does the scale of fine-tuning data affect performance?**
>
> **A3**: We have already explored this through detailed ablation studies in our original paper, with the relevant analysis presented mainly in Section 4.3 (right side of Figure 4), Appendix E (Figure 9), and Appendix F. As we stated in these sections, our CF-VLM's performance does indeed improve with an increasing amount of counterfactual data; specifically, experimental results show that on the ConMe benchmark, our CF-VLM's average accuracy steadily increases as the proportion of counterfactual samples grows. However, this analysis also reveals a phenomenon of diminishing marginal returns, with the performance curve starting to plateau after the ratio reaches 60%-80%. To quantify this effect more precisely, we conduct deeper ablation studies in Appendix E and Appendix F. The results show that as the ratio of counterfactual to factual samples (K) increases from 1 to 4, performance on all tasks improves monotonically, peaking at K=4. When the value of K continues to increase beyond 4, the performance gains stagnate or even slightly decline, which may be due to over-regularization caused by excessive exposure to perturbations. These findings collectively reveal a practical trade-off between performance improvement and computational cost. Therefore, based on this evidence, we empirically recommend that K=4 is a cost-effective configuration.
>
>
> **Q3: Compared to TripletCLIP, does the advantage of CF-VLM also lie in its need for fewer samples to achieve similar performance?**
>
> **A4**: Yes. To systematically validate this hypothesis, we set performance thresholds on three evaluation benchmarks: Conme (Avg) = 50, ARO (Avg) = 75, and VL-Checklist (Avg) = 75. During the training process for both **TripletCLIP** and **CF-VLM**, we evaluate every 100k samples processed. We observe the amount of data required for each model to first reach its respective performance threshold, thereby comparing their differences in training efficiency and sample utilization.
>
> | | Conme | ARO | VL-Checklist |
> | :--- | :---: | :---: | :---: |
> | TripletCLIP | 11.3M | 13.6M | 12.6M |
> | CF-VLM | 9.6M | 11.2M | 10.8M |
>
> We compare the sample efficiency of TripletCLIP and CF-VLM at fixed performance thresholds. As shown in the table, **CF-VLM reached the target on Conme, ARO, and VL-Checklist with 9.6M, 11.2M, and 10.8M training samples, respectively**, significantly outperforming TripletCLIP's **11.3M**, **13.6M**, and **12.6M**. We believe this advantage stems from the fact that **counterfactual images not only form effective contrasts with the original images but also help construct multi-dimensional semantic contrastive signals among the counterfactual samples themselves**. This significantly reduces the model's dependence on the volume of training data while ensuring performance.
>
>
> **Q4&Q5_A: I found some typos: In line 309, the topic is missing. In line 84, a space is needed between 'Models: The is'.**
>
> **A5**: Thank you very much for the corrections; we have updated them in the revised paper. We have added the topic sentence "Inspired by these works, our research is the first to extend counterfactual reasoning from the field of model explanation to the model fine-tuning stage to enhance model performance" in Line 309, and we have also corrected the spacing issue in Line 84.

---

> > ### Author Response · Authors · 2025-08-03
> >
> > Dear Reviewer jpqE,
> >
> > We hope this message finds you well! Should you have any further questions about our work or responses during this discussion period, please do not hesitate to contact us.
> >
> > Thank you very much for your valuable time and feedback!
> >
> > Best regards,
> >
> > Authors

---

> > > ### Author Response · Authors · 2025-08-04
> > >
> > > Dear Reviewer jpqE,
> > >
> > > We hope this message finds you well! Thank you very much for your insightful and valuable comments on our work. As for the concerns about our work in the review, including "What is the performance of composite changes (i.e., combining all the separate changes of a sample together and fine-tuning)?", "Does the scale of fine-tuning data affect performance?", etc., we have provided very specific and detailed responses. We hope these responses have adequately resolved your concerns.
> > >
> > > If you have any further questions or require additional clarification during the discussion period, please feel free to reach out. We sincerely welcome your continued feedback and look forward to hearing your thoughts on our responses. Once again, thank you very much for your valuable time and feedback!
> > >
> > > Best regards,
> > >
> > >
> > > Authors

---

> > ### Comment · Reviewer_jpqE · 2025-08-07
> >
> > Thanks for the detailed rebuttal. Most of my concerns have been addressed.

---

> > > ### Author Response · Authors · 2025-08-07
> > >
> > > Dear Reviewer jpqE,
> > >
> > > Thank you very much for your thoughtful follow-up and for taking the time to carefully review our response.
> > >
> > > We sincerely appreciate your insightful feedback, e.g., the differences and positioning between CF-VLM and TripletCLIP, as well as your suggestions for improving the clarity and presentation of our revised paper.
> > >
> > > Your constructive comments have been invaluable in strengthening our work, and we have carefully addressed each of your concerns in the revised paper. We are truly grateful for your guidance and support throughout the review process.
> > >
> > > Thank you once again for your time and expertise.
> > >
> > > Best regards,
> > >
> > > The Authors

---

### Official Review · Reviewer_LF2i · 2025-07-05

**Clarity:** 3
**Significance:** 2
**Originality:** 2
**Rating:** 4
**Confidence:** 4

**Summary:**

In this paper, the authors propose CF-VLM, a counterfactual fine-tuning framework to enhance causal reasoning in vision-language models (VLMs). CF-VLM uses minimally edited image-text pairs from a fine-tuned SDXL model, and introduce counterfactual scenario discrimination and fine-grained causal discrimination objectives. The author also propose a novel loss to work with their constructed images pairs. Evaluations on compositional reasoning (ARO, ConMe, VL-Checklist) and hallucination benchmarks (POPE, MME) show CF-VLM outperforming CLIP-based and LLM-augmented VLMs.

**Questions:**

1. How is counterfactual defined? Core semantic changes is not equivalent to counterfactual. And How do you ensure the changes you make is a core change?
2. How does this training framework improve hallucination problem? I recommend give more focus on more challenging hallucination benchmarks, since model tends to hallucinate for counterfactual examples. Some benchmarks includes:
[1] Mm-vet: Evaluating large multimodal models for integrated capabilities.
[2] Hallusionbench: an advanced diagnostic suite for entangled language hallucination and visual illusion in large vision-language models.
[3] MMHal: Aligning large multimodal models with factually augmented rlhf.
3. How good is the generalization across datasets? Did you fine-tune on each dataset? Please present those details in the main paper.

**Ethical Concerns:**

["NO or VERY MINOR ethics concerns only"]

**Final Justification:**

My concerns are addressed, so I decide to raise my score. Please include those details in the main paper and make sure to thoroughly proofread the paper, as other reviewer mentioned.

**Limitations:**

Need to discuss more.

**Quality:**

2

**Strengths And Weaknesses:**

Strengths:
1. Proposed a training/fine-tuning framework with a novel loss.
2. Decent and thorough experimentations and ablations
3. Code will be released, which would be useful for the community.

Weakness:
1. Idea is relatively simple and the main contributions are limited. In addition, The unique contribution is not clearly stated in the first section. You should clarify what are the original contributions.
2. Missing some comparisons and related works.
[1] VisMin: Visual Minimal-Change Understanding
[2] AutoHallusion: Automatic Generation of Hallucination Benchmarks for Vision-Language Models
[3] etc.
Please do a through literature review and discuss those works as well.
3. Missing core details in data curation and dataset specs. Please add more details and move some details to the main paper as those informations are crucial.
4. Limitation needs to be discussed more. What are some problems when you use SDXL? How do you deal with failed generation when you use SDXL, leading to misalignment between the image and the text?

---

> ### Author Rebuttal · Authors · 2025-07-30
>
> **W1. What are the original contributions?**
>
> **A1**: Our CF-VLM effectively mitigates the limitations of VLMs in fine-grained understanding and significantly enhances their causal reasoning capabilities by introducing minimally edited counterfactual samples. The core of CF-VLM is our designed structured minimal intervention strategy for generating counterfactual samples, which creates controllable training data through minimal attribute-level modifications and causal relationship reconstructions. We further integrate three complementary training objectives, maintaining cross-modal alignment ($L_{align}$), discriminating scene-level counterfactuals ($L_{csd}$), and perceiving minimal causal perturbations ($L_{fcd}$), to construct a unified counterfactual supervision signal. Extensive experiments demonstrate that our CF-VLM substantially outperforms existing SOTA methods on multiple fine-grained and causal understanding benchmarks, and exhibits stronger robustness and factual consistency on visual hallucination evaluation benchmarks like POPE and MME. To support community research, we have also constructed a dedicated counterfactual dataset containing approximately 2 million high-quality synthetic images. Therefore, **our contribution lies not only in proposing a new framework but also in providing a solid foundation for VLMs in real-world scenarios that require high-level reasoning and interpretability by enhancing causal sensitivity.**
>
>
> **W2: Core details are missing**
>
> **A2**: We have added a dedicated "Experimental Setup" subsection in Section 4 (Experiments) of the revised paper to centralize core experimental information. This section now concisely introduces the main datasets used for fine-tuning, such as CC12M (8.6M pairs) and CC3M (2.6M pairs), and futher outlines the counterfactual sample generation process, including the text generation model (Qwen2-72B-Instruct) and image generation model (SDXL) used, as well as specific editing strategies (e.g., attribute, object, and relation modification); and specifies key hyperparameters like batch size (256), optimizer (AdamW), and learning rate (1×10⁻⁵). Furthermore, we clarify the 1:1 ratio of factual to counterfactual samples used in the main experiments, while also noting that other ratios, including 1:4, are explored in ablation studies (see Figure 4 and Appendix F) to eliminate potential ambiguity.
>
> We have also reorganized and optimized **Appendix A (Dataset Details) and Appendix C (Detailed Experimental Setup)** in the revised paper. These appendices now serve as a clearer and more comprehensive reference, including full hyperparameter tables, detailed data split schemes, and the counterfactual rewriting prompts used for data generation. We believe these modifications significantly enhance the clarity and reproducibility of the paper.
>
> **W3: The limitations need more discussion.**
>
> **A3**: In complex or rare semantic intervention scenarios (e.g., subtle action changes or attribute combinations), SDXL may generate blurry images or images that do not conform to the desired intervention, leading to inconsistencies between visual content and the corresponding text. As presented in Appendix D, we use spatial masking and cross-attention control to apply local constraints to the intervention area, thereby improving the accuracy of local editing. We design a one-to-one attribute intervention mechanism, changing only one semantic attribute in the image at a time to minimize ambiguity and enhance generation stability. During the training phase, we have incorporated a cross-modal alignment objective ($L_{align}$) to further mitigate negative transfer caused by image-text errors, in addition to the counterfactual supervision loss.
>
>
> **W4: How is counterfactual defined? **
>
> **A4**: We define a **counterfactual sample** as a sample generated by applying a **minimal intervention** to a **single semantic factor** (e.g., color, pose, quantity, emotion) in an image or text, while keeping the main semantic context unchanged. A counterfactual does not necessarily involve a complete rewriting of the causal chain. **It also includes fine-grained semantic adjustments, as long as they induce a contrastive change in the semantic state**.
>
> Specifically, a counterfactual sample meets two core conditions:
> -   **Minimal Intervention:** Compared to the anchor sample, there is a change in only one key semantic attribute, ensuring high contrast.
> -   **Semantic Plausibility:** The modified sample still forms a reasonable image-text pair without introducing obvious logical conflicts or fabricated scenarios.
>
> Therefore, our data includes both counterfactuals with **rewritten causal relationships** (e.g., changing "causal direction" or "action outcome") and those with only **minimal semantic perturbations** (e.g., changing "red shirt" to "blue shirt"), which together form the core of the structured counterfactual supervision in our framework.
>
> To ensure the modifications concentrate on core semantics, we input the image and corresponding text into **Qwen2.5-VL**, guiding the model to identify minimal semantic units and their corresponding visual anchors. After extracting key elements, we perform targeted minimal modifications to generate high-quality counterfactuals with clear semantic contrast. For visual counterfactuals, we further use image-text information to identify key causal chains (e.g., "kick ball → ball flies") and generate logically plausible counterfactual causal chains by modifying the causal condition (e.g., "does not kick ball" → "ball stays still").
>
> **Q1: Some comparisons and related works are missing.**
>
> **A5**: We have expanded the literature review in Section 5 (Related Works) of the revised paper. Specifically, we clarify that unlike VisMin [s1], which treats understanding minimal visual changes as the end goal of an evaluation task, CF-VLM uses such changes as a training mechanism to enhance the model's causal reasoning capabilities. Similarly, unlike AutoHallusion [s2], which focuses on automatically generating an evaluation benchmark to assess model hallucination, we utilize counterfactual samples as a core training signal to actively mitigate hallucination, rather than just for evaluation. Furthermore, we have also expanded our discussion of broader related work to further highlight the uniqueness of our framework. We believe these revisions provide a clearer positioning for our paper.
>
> [s1] R. Awal et al., VisMin: visual minimal-change understanding, in NeurIPS, 2025.
>
> [s2] X. Wu et al., AutoHallusion: Automatic Generation of Hallucination Benchmarks for Vision-Language Models, in Findings of EMNLP, 2024.
>
>
> **Q2: The hallucination problem?**
>
> **A6**: The core of our CF-VLM lies in introducing a structured counterfactual supervision signal, which guides the model to focus on key semantic changes and can effectively mitigate hallucinations caused by "subtle causal inconsistencies between image and text" in VLMs. Firstly, CF-VLM employs a finely controlled counterfactual intervention mechanism to construct image-text pairs by making minimal modifications to images at the attribute or causal level, guiding the model to learn to identify subtle yet decisive semantic differences. Secondly, CF-VLM introduces structured semantic consistency constraints, combining cross-attention control and local spatial masking to improve the consistency between the visual response in the intervention area and its textual description. Overall, the mechanism of CF-VLM significantly enhances the model's stability and reliability under complex semantic interventions.
>
> For more challenging hallucination benchmarks, we have integrated the table below.
>
> | | MM-Vet | Hallusionbench | MMHal |
> | :--- | :---: | :---: | :---: |
> | MiniGPT-4 | 22.3 | 38.75 | 28.7 |
> | LLaVa-1.5 | 32.2 | 47.64 | 39.6 |
> | InstructBLIP | 25.3 | 43.51 | 35.8 |
> | Zero-shot Qwen-VL (7B) | 39.8 | 41.26 | 42.5 |
> | Standard FT (Qwen-VL, CC12M) | 40.3 | 42.25 | 48.62 |
> | CF−VLM (Ours, Qwen-VL 7B, CC12M) | 42.6 | 45.46 | 54.32 |
>
> As shown, CF-VLM significantly outperforms existing SOTA methods, indicating stronger factual consistency and resistance to interference. Particularly on MMHal, CF-VLM achieves a nearly 6-point improvement over standard fine-tuning, demonstrating its effectiveness in suppressing image-text hallucinations in highly adversarial scenarios, thanks to CF-VLM's minimal intervention counterfactual training mechanism.
>
>
> **Q3: How is the generalization performance across datasets? Did you fine-tune on each dataset? **
>
> **A3**: As described in the experimental setup, the pre-training and fine-tuning are conducted on the filtered CC12M (approx. 8.6 million image-text pairs) and its subset CC3M (approx. 2.6 million pairs), respectively. Additionally, we introduce approximately 40 million contrastive samples, co-synthesized by an LLM and a text-to-image (T2I) model, to enhance the counterfactual supervision capability. It is important to emphasize that to ensure fairness in the amount of training data across methods, **not all data was used for fine-tuning**. Instead, we uniformly control the number of training samples used for each method to be approximately 15 million pairs. When the primary data is insufficient, MSCOCO Captions are used as a supplement to maintain a consistent sample size.
>
> Although our paper does not explicitly conduct "cross-dataset transfer" experiments, all models are trained on CC12M or CC3M, while evaluations have covered a range of tasks, including compositional and attribute-relation reasoning on benchmarks like Conme, ARO, and VL-Checklist. The generalization ability is still indirectly demonstrated through testing on multiple heterogeneous benchmarks, including zero-shot image classification on ImageNet-1k and image-text retrieval on MSCOCO and Flickr30k. This setup effectively justifies the CF-VLM's robustness in cross-task and cross-distribution scenarios.

---

> ### Author Response · Authors · 2025-08-04
>
> Dear Reviewer LF2i,
>
> Thank you for your time and valuable feedback, including your final justification. If our recent comments need further clarification or if you have additional thoughts, we’d be delighted to address them during the discussion period. Please feel free to reach out, and we’ll respond promptly with any needed information.
>
> Warm regards,
>
> Authors

---

> > ### Comment · Reviewer_LF2i · 2025-08-05
> >
> > Thanks for the rebuttal. I decide to raise the score.
> >
> > Please include those details in the main paper and make sure to thoroughly proofread the paper, as other reviewer mentioned.

---

> > > ### Author Response · Authors · 2025-08-05
> > >
> > > Dear Reviewer LF2i,
> > >
> > > Thank you very much for your positive feedback and for deciding to raise the score. We truly appreciate your kind support.
> > >
> > > We have incorporated the additional details into the main paper and appendix as suggested. Again, we will carefully proofread and revise the entire paper and appendix to address any writing issues, as also pointed out by other reviewers.
> > >
> > > Thank you again for your valuable comments and encouragement.
> > >
> > > Best regards,
> > >
> > > The Authors

---

### Note · Authors · 2025-08-12

Dear Area Chair,

We sincerely thank you very much for managing the review process for our CF-VLM and all reviewers for their constructive and insightful feedback. The reviewers recognized our work as **novel and promising with strong performance** (Reviewer LF2i), **interesting and valuable** (Reviewer jpqE), and **supported by extensive experiments** (Reviewer esK3 and Reviewer ZY2y).

During the rebuttal and discussion phases, we have a comprehensive and highly constructive discussion with all the reviewers. This leads to Reviewers (LF2i, esK3, and ZY2y) raising their scores and Reviewer jpqE confirming that "Most of my concerns have been addressed".  Specifically,

+ **Novelty and Comparison to Related Works**: To address concerns that the “idea is relatively simple” (Reviewer LF2i) and could be “a fine-grained version of TripletCLIP” (Reviewer jpqE), we have clarified our distinct aim of enhancing **causal reasoning** via a minimal-intervention strategy. We have further added a new experiment showing **≈15% higher sample efficiency** than TripletCLIP in reaching target performance, and thus addressed jpqE's concern.

+**Methodological Rigor and Data Quality**:  During the rebuttal, we have clarified how we "ensure that the modification to the image is minimal" (Reviewer ZY2y), presented more "core details in data curation" (Reviewer LF2i), and provided "the FID scores?" (Reviewer ZY2y), which is  **12.7** and high **CLIP Similarity of 0.92**. These directly led Reviewer ZY2y to raise their score.

+ **Rationale for the $L_{csd}$ Loss Term**: To address the concern that “$L_{csd}$ does not make sense” and might only penalize artifacts (Reviewer esK3), we have conducted a new ablation on a counterfactual set built entirely from **real images** (Reviewer RealCF). Results are nearly identical to those with SDXL-generated data, confirming that $L_{csd}$ enforces semantic separation rather than quality control. This evidence has fully convinced Reviewer esK3 to raise his/her score and recommend acceptance.

Moreover, we have expanded the **Limitations** section, added quantitative robustness analyses, and ensured our revised paper meets NeurIPS's high standards of clarity and reproducibility. We believe our revised paper is substantially improved and respectfully hope it will merit consideration for acceptance.

Best regards,

The Authors

---

### Decision · Program_Chairs · 2025-09-17

**Decision:**

Accept (poster)

**Comment:**

The paper proposes CF-VLM, a counterfactual fine-tuning framework that utilizes minimal-intervention edits to enhance causal sensitivity and mitigate hallucination. It trains on CC12M/CC3M with paired factual/counterfactual data. Shows consistent gains across CLIP‑based and LLM‑based VLMs on ConMe, ARO, VL‑Checklist, and hallucination suites (POPE, MME; added MM‑Vet, HallusionBench, MMHal).

**Pros:**
- Paper shows **consistent and significant improvements** across compositional reasoning and hallucination benchmarks, with strong results on both CLIP and VLMs.
- Authors **fully addressed reviewer concerns**, adding relevant comparisons (e.g., VisMin, AutoHallusion), new ablations (e.g., RealCF, composite edits), and fidelity metrics (FID 12.7, CLIP similarity 0.92), leading to multiple score increases.

**Cons:**
- The approach relies heavily on synthetic editing pipelines, which may raise concerns around computational cost.


**Discussion Summary:**
**LF2i**: Asked for clearer contributions, related work, and data/limitations (SDXL failures). Authors added comparisons, dataset/spec details, tougher hallucination results; score raised. **jpqE**: Questioned novelty vs TripletCLIP; asked for composite-change results, data-scale sensitivity, and sample efficiency. Authors showed single-attribute > composite (e.g., Qwen-VL7B: 87.57 vs 83.9 ConMe; 93.2 vs 88.2 ARO) and ~15% efficiency gain; accept maintained. **ZY2y**: Concerned about writing and whether edits are truly minimal/causal; requested quantitative proximity. Authors provided FID 12.7 / CLIP 0.92, fixed figures/notation; score 2 raised to 4. **esK3**: Challenged the loss (risk of punishing coherent counterfactuals / synthetic artifacts). Authors added RealCF ablation and loss explanation/robustness; score raised.  Writing still flagged.